# Two stages of bandwidth scaling drives efficient neural coding of natural sounds

**Fengrong He[1], Ian H. Stevenson[1,2,3], Monty A. Escabí[1,2,3,4]** *

**1** Biomedical Engineering, University of Connecticut, Storrs, Connecticut, United States of America, **2** Psychological Sciences, University of Connecticut, Storrs, Connecticut, United States of America, **3** The Connecticut Institute for Brain and Cognitive Sciences, University of Connecticut, Storrs, Connecticut, United States of America, **4** Electrical and Computer Engineering, University of Connecticut, Storrs, Connecticut, United States of America

* escabi@engr.uconn.edu

**Data Availability Statement:** Auditory model is available via GitHub (https://doi.org/10.5281/zenodo.7245908).

**Funding:** This work was supported by the National Institute On Deafness And Other Communication

## Abstract

Theories of efficient coding propose that the auditory system is optimized for the statistical structure of natural sounds, yet the transformations underlying optimal acoustic representations are not well understood. Using a database of natural sounds including human speech and a physiologically-inspired auditory model, we explore the consequences of peripheral (cochlear) and mid-level (auditory midbrain) filter tuning transformations on the representation of natural sound spectra and modulation statistics. Whereas Fourier-based sound decompositions have constant time-frequency resolution at all frequencies, cochlear and auditory midbrain filters bandwidths increase proportional to the filter center frequency. This form of *bandwidth scaling* produces a systematic decrease in spectral resolution and increase in temporal resolution with increasing frequency. Here we demonstrate that cochlear bandwidth scaling produces a frequency-dependent gain that counteracts the tendency of natural sound power to decrease with frequency, resulting in a whitened output representation. Similarly, bandwidth scaling in mid-level auditory filters further enhances the representation of natural sounds by producing a whitened modulation power spectrum (MPS) with higher modulation entropy than both the cochlear outputs and the conventional Fourier MPS. These findings suggest that the tuning characteristics of the peripheral and mid-level auditory system together produce a whitened output representation in three dimensions (frequency, temporal and spectral modulation) that reduces redundancies and allows for a more efficient use of neural resources. This hierarchical multi-stage tuning strategy is thus likely optimized to extract available information and may underlies perceptual sensitivity to natural sounds.

## Author summary

Theory suggests that the auditory system evolved to optimally encode salient structure in natural sounds—maximizing perceptual capabilities while minimizing metabolic demands. Here, using a multi-stage model of the auditory system and a collection of environmental sounds, including vocalizations such as speech, we demonstrate how auditory

Disorders of the National Institutes of Health (R01DC015138, M.A.E. and I.H.S; R01DC020097, M.A.E. and I.H.S) and National Science Foundation (2043903, M.A.E.). The content is solely the responsibility of the authors and does not necessarily represent the official views of the NIH or NSF. The funders had no role in study design, data collection and analysis, decision to publish, or preparation of the manuscript.

**Competing interests:** The authors have declared that no competing interests exist.

responses may be optimized for equalizing the power distribution of natural sounds at two levels. This processing strategy may improve the allocation of resources throughout the auditory pathway, while ensuring that a broad range of auditory features can be detected and perceived. Such a multi-stage strategy for processing natural sounds likely contributes to human perceptual capabilities and adopting such a code could enhance the performance of auditory prosthetics and machine systems for sound recognition.

## Introduction

The cochlea decomposes sounds into distinct frequency channels and produces patterned fluctuations or *modulations* across both time and frequency that serve as input to the central auditory system. For natural sounds, these spectro-temporal modulations are not uniformly distributed, but encompass a limited set of all possible sound patterns [1,2], much as natural images encompass a restricted subset of visual patterns [3,4]. After being transmitted out of the cochlea and along the auditory nerve, modulations in the envelope of natural sounds are further decomposed by the central auditory system, where neurons in mid-level structures such as the auditory midbrain (inferior colliculus) are selectively tuned for a unique subset of spectro-temporal modulations [5–7]. This secondary decomposition into modulation components resembles the modulation power spectrum (MPS) analysis that has been used to characterize and to identify salient features in natural sounds [1,2,8,9].

Both spectral and temporal modulations in the envelope of speech and other natural sounds are perceptually salient cues that are critical for perception and recognition of sounds [1,10]. Temporal modulations in natural sounds can span several orders of magnitude. Relatively slow temporal fluctuations in the rhythm range (<25 Hz), for instance, are critical for parsing speech and vocalizations sequences and for musical rhythm perception [11,12]. Intermediate temporal modulations (~50–100 Hz) contribute to the perception of roughness, and the fastest temporal modulations (~80–1000 Hz) contribute to perceived pitch [13,14]. Similarly, in the frequency domain, spectral modulations also convey critical information about the sound content and can contribute to timbre and pitch perception [1,15]. In speech, for instance, harmonic structure created by vocal fold vibration during voiced speech generates high-resolution spectral modulations (resolved harmonics) that can indicate voice quality, gender identity, and overall voice pitch [1]. On the other hand, resonances generated by the postural configuration of the vocal tract produce broader spectral modulations (e.g., formants) that can contribute towards the identity of vowels [15]. Evidence also suggests that spectral modulations contribute towards the perception of timbre in music and are critical for instrument identification [16,17].

How the auditory system extracts and utilizes spectral and temporal modulations and how neural computations contribute towards basic perceptual tasks is not well understood. Following the efficient coding hypothesis originally proposed by Barlow for visual coding [18], it's plausible that the auditory filter computations are optimized to efficiently encode and extract available information in the envelope of natural sounds. Indeed, several studies have shown that spectral and temporal modulations in natural sounds are highly structured [2,8] and that neural tuning properties at various stages of the auditory pathway appear to be optimized to extract available acoustic information [8,19–24]. Using a generative encoding model, the optimal frequency decomposition of natural sounds resembles a cochlear decomposition in which the filter tuning exhibits bandwidth scaling, that is, bandwidths increase proportional to the filter best frequency [23,25]. Thus, the initial decomposition might be optimized to extract and

represent available information in natural sounds. Similarly, the second-order decomposition of sounds into spectro-temporal modulation components observed in the auditory midbrain is predicted by computational models designed to optimally encode spectrographic information with a sparse representation [24]. Once again, as for the cochlear filters, auditory modulation filters perform a multi-scale decomposition, but do so with respect to the second-order sound modulations. Both the spectral and temporal modulation filter bandwidths for this scheme scale proportional to the modulation frequency of each filter. Intriguingly, modulation filter bandwidth scaling has been observed physiologically [8] and is also consistent with human perception of modulated sounds [26,27].

Although the bandwidth scaling characteristics of peripheral (carrier decomposition) and mid-level (modulation decomposition) auditory pathway tuning are well described physiologically, the consequences of this dual tuning strategy, both computationally and perceptually, are not fully understood. In particular, it is unclear why bandwidth scaling is evident in peripheral and mid-level auditory structures and how it impacts auditory feature representations for natural sounds. We demonstrate that, in contrast to widely used Fourier sound decompositions which preserve the original power distribution of natural sounds, the scaling characteristics of the peripheral and mid-level auditory filters serve to whiten the neural outputs of the cochlea and midbrain, and hence, increase the available entropy in natural sounds. This dual-tuning strategy is consistent with efficient coding principles and provides a normative framework for understanding perception of natural sounds.

## Methods

### Natural sound ensembles and analysis

To study the role of auditory filter tuning and the neural transformations for representing natural sounds, we analyzed the modulation statistics of natural sound ensembles using a physiologically-inspired auditory model. The model consists of a peripheral filterbank stage that models the initial, cochlear decomposition of a sound waveform into spectro-temporal components. A second mid-level modulation filterbank stage decomposes the cochlear spectrogram of each sound into modulation components and is inspired by the modulation decomposition thought to occur in the auditory midbrain [28,29] (Fig 1). Both the peripheral and mid-level model filters are designed to match tuning characteristics observed physiologically and perceptually [8,26,27]. For comparison, we also analyze natural sounds using Fourier-based spectrographic and modulation decompositions widely used for sound analysis, synthesis, and sound recognition applications. All of the models were implemented in MATLAB and are available via GitHub (https://doi.org/10.5281/zenodo.7245908).

The selected sounds were chosen to represent two broad classes of sounds: background environmental sounds and animal vocalizations. Sounds within each category were divided into sub-categories representing the specific source of the background sound or the species generating the vocalization. In all, we analyzed 29 sound categories, including 10 background sound categories, 18 vocalization categories and white noise as a reference. Example natural background sound categories included crackling fire, running water, and wind, while vocalization categories included human, parrot, and new world monkey speech/vocalizations. Each category contained 3 to 60 sound recordings lasting between 5 seconds and 203.8 seconds (average = 38.1s). The length of each recording was limited by the recorded media, but we required a total minimum category length of 90 seconds for each category to assure that sufficient averaging could be performed to adequately assess the modulation statistics. In total, we analyzed 457 sound segments totaling 4.8 hours of recording. All sounds were sampled at 44.1kHz. The complete list of the sound categories and media sources is provided in S1 Table and S1 Text.

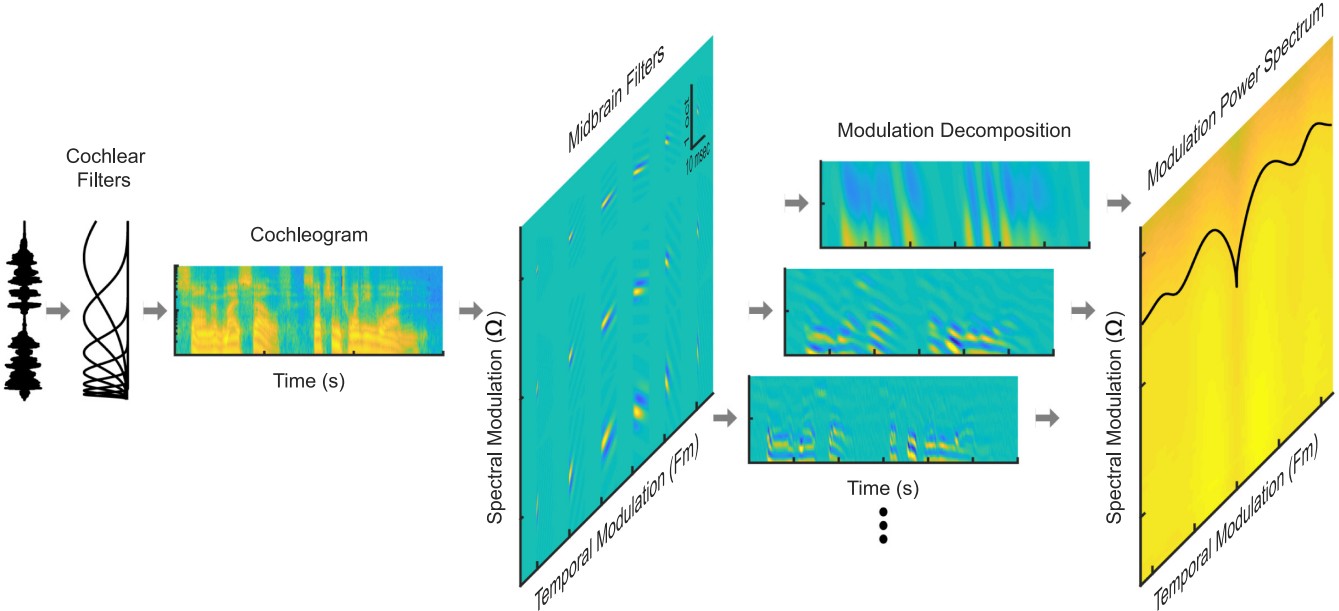

**Fig 1. Using a multi-stage auditory system model to measure the modulation power spectrum of natural sounds.** A cochlear filterbank stage first decomposes the sound pressure waveform (show for speech) into a spectro-temporal output representation (cochleogram). The cochleogram is then decomposed into modulation bands by a bank of spectro-temporal receptive fields (STRFs) of varying resolution modeled after the principal auditory midbrain nucleus (inferior colliculus). The resulting multi-dimensional output represents the sounds in frequency, time, temporal modulation, and spectral modulation. The modulation power spectrum (MPS), as measured through this auditory midbrain-inspired representation, is generated by measuring and plotting the power in each of the modulation band outputs versus temporal and spectral modulation frequency.

## Auditory model decomposition

**Cochlear spectrogram.** The first stage of auditory model consists of a peripheral filter-bank that models the frequency decomposition and envelope extraction performed by the cochlea. The resulting output, referred to as the *cochlear spectrogram* or *cochleogram*, captures the spectro-temporal modulations of the sound as represented through the cochlear model. The sound waveform, $s(t)$, is first convolved with a set of $N = 664$ tonotopically arranged gamma-tone filters

$$s_k(t) = h_k(t) * s(t) \tag{1}$$

with impulse response

$$h_k(t) = A \cdot t^{n-1} \cdot cos(2\pi f_k t) \cdot e^{-2\pi \cdot b(f_k) \cdot t} u(t) \tag{2}$$

where $f_k$ represents the $k^{\text{th}}$ filter characteristic frequency (CF), $b(f_k)$ is the filter bandwidth, $u(t)$ is the unit step function, $*$ is the convolution operator and the filter gain, $A$, is selected to achieve unity maximum gain in the frequency domain. The filter characteristic frequencies (CF) are ordered logarithmically between 100 Hz and 10 kHz (0.01 octave spacing) to model the approximate logarithmic position vs. frequency relationship in the cochlea [30,31]. Furthermore, bandwidths scale according to

$$b(f) = 24.7 \cdot \left( 4.37 \cdot \frac{f}{1000} + 1 \right) \text{ Hz}, \tag{3}$$

such that bandwidths increase with filter CF [32,33]. Next, we computed the Hilbert transform

magnitude to extract the envelope of each channel

$$e_k(t) = |s_k(t) + i \cdot H\{s_k(t)\}|, \tag{4}$$

where $H\{\cdot\}$ is the Hilbert transform operator and $i = \sqrt{-1}$. Finally, to account for the fact that synaptic filtering at the hair-cell synapse limits the temporal synchronization and modulation sensitivity of auditory nerve fibers [34,35], the final cochlear outputs were derived by convolving the impulse response of a synaptic lowpass filter ($h_{synapse}(t)$) with the sound envelope of each cochlear channel ($e_k(t)$)

$$S_C(t, x_k) = e_k(t) * h_{synapse}(t) \tag{5}$$

where $x_k = log_2(f_k/100)$ is the frequency in units of octaves above 100 Hz for the k-th filter channel. For each of the cochlear channels, this synaptic filter is modeled as a B-spline lowpass filter with a lowpass cutoff frequency of 750 Hz. Altogether, $S_C(t, x_k)$, provides a decomposition of the original sound in terms of spectro-temporal modulations using a filterbank model of the auditory periphery.

**Mid-Level modulation decomposition.** Following the peripheral cochlear decomposition, we use a mid-level filterbank to extract spectral and temporal modulations in the cochleogram. In this second stage of the model, the cochlear spectrograms are passed through a multiresolution bank of two-dimensional filters designed to model spectro-temporal receptive fields (STRFs) in the auditory midbrain. Here, STRFs contain both excitatory and inhibitory (or suppressive) integration components and STRF filters are designed to match the tuning properties reported physiologically in the inferior colliculus [8]. The STRF filters are modeled using a Gabor-alpha function that captures the structure of auditory midbrain receptive fields [36]:

$$STRF(t, x; f_{mo}, \Omega_o)$$

$$= A_m \cdot \frac{t}{\tau} \cdot e^{-\left(\frac{t-\tau}{\tau}\right)} e^{\left(-2 \cdot \frac{x^2}{bw^2}\right)} \cos(2\pi\Omega_0 x + 2\pi f_{m0} t + \phi) u(t), \tag{6}$$

where $f_{m0}$ and $\Omega_0$ are the best temporal and spectral modulation frequency parameters of each individual STRF, respectively. These primary receptive field parameters determine the modulation tuning of each model neuron and are varied systematically on an octave scale between $f_{m0}$ = -512 to 512 Hz (0.25 octave steps) and $\Omega_0$ = 0.1 to 3.6 cycles/oct (0.1 octave steps). We choose octave spacing for these primary receptive field parameters because both mapping and modulation processing studies [28,29,37] indicate that modulation preferences are roughly evenly distributed when plotted on an octave scale. Secondary receptive field parameters include the receptive field phase ($\phi$, which accounts for the alignment of excitation and inhibition), the temporal receptive field decay time-constant ($\tau$, which determines the temporal duration of the STRF) and the spectral bandwidth ($bw$, which determines the spectral spread of the STRF in octaves). These secondary parameters are selected based on physiologically measured trends for inferior colliculus that are described subsequently (*Selecting Physiologically Plausible Modulation Tuning Parameters*). Finally, the receptive field amplitude,

$$A_m = \frac{4}{\sqrt{\pi} \cdot e} \cdot \frac{1}{bw \cdot \tau^2} \tag{7}$$

is selected so that the filters have a constant peak gain of 1 in the modulation domain.

The mid-level modulation filterbank output to a particular sound, $S_M$, is obtained by convolving the model STRFs with the sound cochleogram according to

$$S_M(t, x, f_{m0}, \Omega_0) = STRF(t, x; f_{m0}, \Omega_0) ** S_C(t, x) \tag{8}$$

where $**$ is a two-dimensional convolution operator (across time and frequency). This operation decomposes the cochleogram into different modulation resolutions determined by the model STRFs (Fig 1). This decomposition is conceptually similar to the cortical decomposition of sounds into spectro-temporal modulation components [10], although in this case, the decomposition accounts for substantially faster temporal modulations and is designed to capture physiological distributions and receptive field characteristics of the auditory midbrain [5,8,36].

**Selecting physiologically plausible modulation tuning parameters.** While the peripheral decomposition of sounds by the cochlea selectively filters the frequency content of natural sounds, the secondary decomposition performed by the auditory midbrain selectively extracts and filters the modulation content. Physiologically, the measured modulation filters have a quality factor of ~1 ($Q$, defined in the modulation domain: the ratio of best modulation frequency to modulation bandwidth; $Q_{f_m} = f_{m0}/BW_{f_m}$; $Q_\Omega = \Omega_0/BW_\Omega$), such that modulation bandwidths *scale* proportional to the best modulation frequencies [8]. Similarly, human modulation bandwidths, which are derived using perceptual measurements, also scale with modulation frequency [26,27].

To match these physiological observations, we set the temporal modulation bandwidths equal to the best temporal modulation frequency ($BW_{f_m} = f_{m0}$) and set spectral modulation bandwidths equal to the spectral modulation frequencies ($BW_\Omega = \Omega_0$). As observed physiologically these modulation domain parameters ($BW_{f_m}$ and $BW_\Omega$) are intimately related to the STRF parameters ($\tau$ and $bw$) [5,8,36], and for the model STRF of Eq 6 it can be shown that:

$$\tau = \frac{\sqrt{\sqrt{2} - 1}}{\pi} \cdot \frac{1}{f_{m0}} \tag{9}$$

and

$$bw = 2 \frac{\sqrt{2 \cdot ln(2)}}{\pi \cdot \Omega_0} \tag{10}$$

(Proofs in S1 Text). Collectively, by combining Eqs 6, 9 and 10, the model STRFs exhibit tuning profiles that follow trends in auditory midbrain and perceptual measurements where the spectro-temporal modulation bandwidths scale with best modulation frequencies.

## Fourier spectrographic decomposition

In addition to decomposing natural sounds through a physiologically inspired auditory model, we also decomposed sounds through a conventional Fourier-based spectrographic decomposition (i.e., short-term Fourier transform). Here the modulations of natural sounds are extracted using a short-term Fourier transform, equivalent to using a constant resolution Gabor filterbank. Although both the Fourier and cochlear spectrogram representations describe the spectro-temporal envelopes of natural sounds, each decomposition uses a unique set of filters with different time-frequency resolution patterns (constant resolution for the Fourier spectrogram versus approximately proportional resolution for the cochleogram) thus yielding uniquely different spectro-temporal representations.

For each sound, the spectrographic representation is given by taking short-term Fourier transform

$$s(t,f) = \int s(\tau)w(t-\tau)e^{-j2\pi f\tau}d\tau \tag{11}$$

and computing the envelope magnitude: $S(t, f) = |s(t, f)|$. In the above, $w(t)$, is a Gaussian window with standard deviation $\sigma$ that localizes the sound in time to the vicinity of $t$ prior to computing the Fourier transform. Alternately, the short-term Fourier transform can be viewed as a filterbank decomposition in which complex Gabor filters of the form $w(t-\tau)\,e^{-j2\pi f\tau}$ are convolved with the stimulus, $s(\tau)$ [38]. The filters have center frequency $f$ and a constant bandwidth that is inversely related to $\sigma$ [38]. This follows from the uncertainty principle which dictates that the spectral and temporal resolution of a filter are inversely related as described below.

## Spectro-temporal resolution and uncertainty principle

To characterize the time-frequency resolution of the cochlear and Gabor filterbanks and to subsequently characterize the structure of the resulting spectro-temporal decompositions, we measured the temporal and spectral resolution of each filter for both filterbanks. The uncertainty principle requires that

$$\sigma_t \cdot \sigma_f \geq 1/4\pi, \tag{12}$$

where equality holds for the Gabor filter case [38]. Here $\sigma_t^2$ and $\sigma_f^2$ are the normalized second-order moments of the filter impulse response and transfer function, respectively. That is, the product of the temporal and spectral resolutions is bounded, and there is a tradeoff between the two in the limiting case where the filter approaches the theoretical best resolution (i.e., for Gabor filters). Conceptually $2\sigma_t$ and $2\sigma_f$ can be thought of as the average *integration time* and *bandwidth* of the filter, which we define as

$$\Delta t = 2 \cdot \sigma_t$$

$$\Delta f = 2 \cdot \sigma_f. \tag{13}$$

The uncertainty principle can then be expressed as

$$\Delta t \cdot \Delta f \geq 1/\pi. \tag{14}$$

For this study, we choose and characterized natural sounds using Gabor filters with three distinct spectro-temporal resolutions: integration times of $\Delta t$ = 10.6, 2.7, and 0.66 ms and corresponding bandwidths of $\Delta f$ = 30, 120, and 480 Hz (36, 141, 567 Hz 3 dB bandwidths, respectively). These decompositions have the same $\Delta t \cdot \Delta f$ resolution and are comparable to those used previously to analyze modulation spectra of speech and other natural sounds [1,2].

## Modulation power spectrum (MPS)

We are broadly interested in understanding how natural sounds are transformed by cochlear and mid-level filters and in determining to what extent auditory filters represent spectro-temporal modulations of natural sounds efficiently. Here we propose to use the modulation power spectrum (MPS) to evaluate representations of spectro-temporal modulations. Conceptually, the MPS is analogous to a power spectrum, but calculated for spectro-temporal modulations of the sound [2]. Since modulations are determined by the filterbank model used for the

spectrographic decomposition [38], the MPS can differ substantially between spectrographic and cochleographic representations [2,8]. As we will also demonstrate, the MPS of a sound is also highly dependent on the modulation filters used to estimate the MPS itself. Here we describe the calculation of the MPS at multiple levels of the auditory processing using the 1) cochlear model and a 2) midbrain model decomposition as well as for the reference 3) Fourier based spectrographic decomposition.

**Fourier spectrogram MPS.** Conceptually, computing the MPS of natural sounds entails measuring the output power through a bank of modulation filters that decompose the sound into isolated modulation components [2]. This can be achieved by taking the two-dimensional Fourier transform of the spectrographic representation and subsequently computing the squared magnitude

$$MPS(f_m, \Omega) = |\iint S(t,x)e^{-j2\pi(\Omega x + f_m t)}dtdx|^2. \tag{15}$$

Conceptually, the two-dimensional Fourier transform of the spectrogram transforms the time and frequency dimensions ($t$ and $x$) into the corresponding temporal and spectral modulation frequencies ($f_m$ and $\Omega$), while the squaring operation is needed to compute the power of each modulation component. Here, due to the limited data size, we used a Welch's periodogram averaging procedure to approximate Eq 15, similar to previous methods [8]. The sound spectrogram, $S(t,x)$, is first partitioned in time into $N$ adjacent non-overlapping segments, $S_n(t,x)$ ($n = 1\cdots N$). The *MPS* is then given by

$$MPS_F(f_m, \Omega) = \frac{1}{N}\sum_{n=1}^{N}|\iint S_n(t,x)w(t-t_n,x)e^{-j2\pi(\Omega x + f_m t)}dtdx|^2. \tag{16}$$

Here, the two-dimensional modulation filters ($w(t-t_n,x)e^{-j2\pi(\Omega x + f_m t)}$) have constant modulation resolution. That is, the estimated power for each modulation frequency component can be viewed as the power that is measured through the corresponding modulation filter. We use a 1.5 seconds duration two-dimensional Kaiser window ($\beta = 3.4$) spanning the full frequency range (0.1 – 10kHz), which in the modulation domain have a constant resolution of 0.8 Hz and 0.1 cycles/kHz (3 dB Bandwidths). Although this procedure differs slightly from the approached originally used by Singh and Theunissen [2], it is theoretically equivalent and, for speech, produces very similar MPS [1].

**Cochlear spectrogram MPS.** Next, to characterize the modulations represented by a cochlear model decomposition, we estimate the MPS of the cochleogram [8]. Due to the bandwidth scaling, nonlinearity (Hilbert transform) and synaptic lowpass filter of the cochlear filters, the cochleogram representation differs from the Fourier spectrogram. However, the cochlear MPS is computed similarly

$$MPS_c(f_m, \Omega) = \frac{1}{N}\sum_{n=1}^{N}|\iint S_{c,n}(t,x)w(t-t_n,x)e^{-j2\pi(\Omega x + f_m t)dtdx}|^2 \tag{17}$$

where $S_{C,n}(t,x)$ denotes the segmented cochlear spectrogram and replaces the Fourier version ($S_n(t,x)$), but the same window is applied to both (see above). Although both the Fourier *MPS* and cochlear $MPS_C$ quantify temporal and spectral modulations, Fourier filters have Hz spacing with constant bandwidth, while cochlear filters have octave spacing with proportional bandwidths. Thus, while temporal modulation frequencies ($f_m$) have a common unit of Hz, the units for spectral modulations frequencies ($\Omega$) differ for the two representations: cycles/octave for the cochlear and cycles/kHz for the Fourier spectrograms.

**Mid-Level / Midbrain Model MPS.**   In the Fourier and cochlear MPS, the spectro-temporal filters have constant spectro-temporal resolution in the modulation domain and can be viewed as a basis set from which arbitrary Fourier and cochlear spectrograms can be synthesized. Here to model the auditory midbrain and to derive a MPS representation of the midbrain model output we consider an alternative decomposition of the cochlear spectrogram. Unlike the cochlear MPS which uses equal-resolution modulation filters, we use model receptive fields based on auditory midbrain STRFs [8,36]. These spectro-temporal receptive fields *scale* in the modulation domain thus resembling scaling observed physiologically [8]. This scaling generates a decomposition analogous to a two-dimensional wavelet decomposition of the cochlear spectrograms. Here the mid-level model MPS is given by the power at the output of the mid-level filterbank:

$$MPS_M(f_m, \Omega) = \int \int S_M(t, x; f_m, \Omega)^2 dt dx, \qquad (18)$$

where $S_M(t, x; f_m, \Omega)$ is the mid-level or midbrain filterbank output (Eq 8). Applying Parseval's theorem and combining with Eq 8, the midbrain MPS can alternately be computed directly in the modulation domain by integrating the cochlear modulation power spectrum ($MPS_C$)

$$MPS_M(f_m, \Omega) = \iint |MTF(\zeta, \gamma; f_m, \Omega)|^2 \cdot MPS_C(\zeta, \gamma) d\zeta d\gamma. \qquad (19)$$

where the modulation transfer function ($MTF(\zeta, \gamma; f_m, \Omega)$) is obtained by taking the Fourier transform of each STRF (see S1 Text). That is, the mid-level MPS is a transformed version of the cochlear model MPS. Here the midbrain model *MTF* magnitudes shape the MPS output of each modulation filter, and the total power for each filter is derived by integrating across spectral and temporal modulation frequencies. Spectral and temporal modulation frequencies in $MPS_M$ share the same units as $MPS_C$ (Hz and cycles/oct). However, because the modulation filters scale with modulation frequency, both $f_m$ and $\Omega$ are now ordered logarithmically.

## Spectral and modulation entropy

Here we use Shannon entropy [39] to characterize the effectiveness of a spectro-temporal decomposition model for encoding natural sounds. Entropy is a metric of waveform diversity and thus serves as a measure of potential information that may be transmitted within a signal coding framework. Here we extend the conventional definition by quantifying the average entropy in the neural response distribution in the frequency or modulation domains. Within this multi-dimensional signal encoding framework, high spectral or modulation entropy indicates that the encoded signal uniformly spans the basis set (i.e., the filters), as might be expected for white noise being represented by conventional Fourier decomposition. Thus, from a neural coding perspective, a signal with high spectral or modulation entropy is expected whenever a sensory signal broadly and uniformly activates all of the neurons in the encoding ensemble [18]. That is, a signal with high entropy is "whitened" by the particular filterbank scheme. Here we measure the entropy associated with the spectral and modulation content of natural sound as represented through the 1) Fourier based, 2) cochlear model, and 3) midbrain model decompositions.

**Spectral entropy.**   For each of the natural sound ensembles and both their Fourier and cochlear model decompositions, we measured and compared the *spectral entropy* [40] as a measure of the efficiency of the spectral decomposition. For a set of *N* spectral decomposition filters, the spectral entropy of a sound is defined by the average expected uncertainty across all filters. The spectral entropy calculation first involves calculating the power spectrum of a

sound, which for the Fourier and cochlear models can be derived by averaging the sampled time dimension in the spectrographic representations as follows:

$$P_{ss}(f_l) = \frac{1}{K}\sum_k |S(t_k, f_l)|^2 \tag{20}$$

$t_k$ is the k-th time sample, $f_l$ is the l-th frequency channel, and $K$ is the number of temporal spectrogram samples. Next, the power spectrum is normalized for unit sum

$$\bar{P}_{ss}(f_l) = \frac{P_{ss}(f_l)}{\sum_{n=1}^{N} P_{ss}(f_n)} \tag{21}$$

so that the normalized power spectrum ($\bar{P}_{ss}(f_l)$) can be treated as a probability distribution (sum of 1). The raw entropies associated with the power spectrum of a sound are then computed as:

$$H = -\sum_{n=1}^{N} \bar{P}_{ss}(f_n) \cdot log_2[\bar{P}_{ss}(fn)] \tag{22}$$

For both the Fourier and cochlear representations, the power spectrum and resulting entropy was estimated for frequencies between 100 Hz and 10 kHz. Furthermore, to allow for comparisons across the different model representations (Fourier vs. cochlear), we consider the maximum possible entropy that can be attained by each filterbank or, equivalently, the *capacity* of the spectral decomposition model as a reference benchmark. The model capacity is achieved when the resulting sound spectrum has a uniform power spectral density (i.e., flat so that $\bar{P}_{ss}(f_n) = 1/N$) and thus a total entropy of $log_2N$. The *spectral entropy* is then defined as:

$$H_s = \frac{H}{log_2N} = -\sum_{n=1}^{N} \frac{\bar{P}_{ss}(f_n) \cdot log_2[P_{ss}(f_n)]}{log_2N} \tag{23}$$

where the entropy is normalized by the theoretical maximum entropy that can be achieved given $N$ decomposition filters. Note that the unnormalized entropy (Eq 22) grows proportional to the number of filters ($N$, which differs for the cochlear and spectrographic decompositions) and thus the entropy is normalized in Eq 23 to remove this dependency. This assures that comparisons can be made across spectral decomposition models with different number of decomposition filters. Spectral entropy can thus be viewed as the fractional entropy that can be achieved by each filter relative to the maximum that is theoretically attainable and thus can be thought of as a measure of efficiency in the population representation. Representations that are more efficient, will activate all of the neural filters uniformly while those that are less efficient will activate a subset of filters more strongly than others.

**Modulation entropy.** We also estimate the entropy associated with the modulation content of each sound. *Modulation entropy* is similar in concept to the spectral entropy described above, but is generalized to two-dimensions

$$H_M = -\sum_i\sum_j \frac{\bar{MPS}(f_{m,i}, \Omega_j) \cdot log_2[\bar{MPS}(f_{m,i}, \Omega_j)]}{log_2(L \cdot M)} \tag{24}$$

where $L$ and $M$ are the number of spectral and temporal filter channels, respectively, $f_{m,i}$ is the $i$th temporal modulation frequency, $\Omega_j$ is the $j$th spectral modulation frequency, and $\bar{MPS}(f_{m,i}, \Omega_j)$ is the normalized MPS (for unit sum). As for the spectral entropy, $log_2(L \cdot M)$ is the theoretical maximum entropy that can be achieved given $L \cdot M$ modulation filter outputs. Thus, values of $H_M$ near 1 would be near the theoretical maximum, indicating an efficient

modulation representation. To characterize how temporal and spectral modulation representations are individually influenced by each of the model decomposition, we also computed the spectral ($H_{SM}$) and temporal ($H_{TM}$) modulation entropy separately

$$H_{SM} = -\sum_j \frac{\bar{MPS}(\Omega_j) \cdot log_2[\bar{MPS}(\Omega_j)]}{log_2 L} \tag{25}$$

$$H_{TM} = -\sum_i \frac{\bar{MPS}(f_{m,i}) \cdot log_2[\bar{MPS}(f_{m,i})]}{log_2 M} \tag{26}$$

where and $\bar{MPS}(f_{m,i})$ and $\bar{MPS}(\Omega_j)$ are the temporal and spectral MPS marginal distributions, respectively.

The total, spectral, and temporal modulation entropy was derived for each sound in each ensemble, as decomposed through a 1) Fourier-based representation, 2) a cochlear model, and 3) a midbrain model. Because the range of spectro-temporal modulations generated by each model are different due to the filterbank characteristics, we use white noise to determine a suitable range of modulation frequencies over which to calculate entropy. Here the range of spectral and temporal modulations used for the entropy calculation were determined by the 90[th] percent power contour of the white noise $MPS$ and $MPS_C$. This ensures that the entropy calculation is performed using modulations that obey the uncertainty principle and that can be reliably identified under each decomposition. Finally, since auditory midbrain neural responses to spectro-temporal modulation are largely limited to less than 500 Hz and 4 cycles/octave [5], and both of these values were less than the upper limit for white noise, we used these values as upper limits for the midbrain representation. This upper limit for the auditory midbrain representation did not bias the entropy calculation, since all representations have a comparable entropy for white noise (Fourier: 0.95, 0.95, 0.95 for $\Delta f$ = 30, 120 and 480 Hz, respectively; Cochlear: 0.95 bits; and Midbrain: 0.93).

## Results

Here we examine how auditory filter transformations influence the neural representations of natural sounds by characterizing the spectrum and modulation statistics of cochlea- and midbrain-inspired sound decompositions. Our full, biologically-inspired auditory model consists of a peripheral set of frequency-selective filters and a subsequent bank of mid-level modulation-selective filters that model the tuning characteristics observed in the cochlea and auditory midbrain, respectively (Fig 1). By comparing biologically-inspired representations to Fourier-based spectrographic decompositions, we demonstrate how peripheral and mid-level auditory filter tuning are better-matched to the statistics of natural sounds ensembles. Together peripheral and midbrain transformations appear to produce a near-optimal, whitened neural representation of the spectro-temporal modulations that are present in natural sounds.

### Tradeoffs in time-frequency filtering resolution and the implications for spectrographic representation of natural sounds

We first examine the consequences of peripheral filter tuning using a cochlear model representation and compare the results to a Fourier-based filter representation. Here we examine natural sounds selected from 28 distinct sound ensembles with a wide range of spectro-temporal characteristics, including animal vocalizations (18) and environmental background sounds (10). Sounds included speech, parrot, and non-human primate vocalizations, for example, as well as, sounds from running water, wind, and crowd noise as backgrounds.

Although both cochlear model and Fourier-based decompositions provide representations of spectro-temporal modulations, they use different filters with distinct impulse response and transfer functions (Fig 2). In the frequency domain, cochlear model filters have bandwidths that scale with frequency; that is, bandwidths vary and increase approximately proportional to the filter best frequency (Fig 2A). At low frequencies, the filters have narrow frequency tuning (in kHz) and thus relatively high spectral resolution while at high frequencies they are broader and less resolved in frequency. In Fig 2A, the cochlear model filters are depicted in log-frequency axis which demonstrates that the filters have approximately equal *proportional* resolution for frequencies above ~1kHz (i.e., ~constant octave bandwidth). In addition, the Gammatone cochlear filters have shallow low-frequency and sharp high-frequency roll-offs that also mirrors the selectivity of auditory nerve fibers [41]. In the time domain, the impulse responses of the cochlear filters differ substantially in their temporal characteristics and amplitudes for different best frequencies (Fig 2C, bottom). As illustrated for three selected filters, low frequency filters have long delays and coarser temporal resolution (note the logarithmic delay axis) while high frequency filters have substantially shorter delays and higher temporal resolution (Fig 2C top, amplitude normalized in order to highlight their temporal characteristics). For example, the filter at 100 Hz has a temporal resolution of $\Delta t$ = 45 ms and group delay of 11 ms indicating that it has relatively poor temporal acuity while the 10 kHz filter has a $\Delta t$ = 1.7 ms and group delay of 0.4 ms which indicates that it responds substantially faster and can synchronize to substantially higher temporal components in the sound. Finally, the peak amplitudes of the filter impulse response (Fig 2C, bottom) increase with increasing frequency which compensate for the bandwidth dependency shown in Fig 2A.

While the cochlear filters have spectral and temporal characteristics that vary in a frequency dependent manner, the Fourier spectrographic decomposition (Fig 2B) uses linearly-spaced filters with constant spectral bandwidth ($\Delta f$). Although the impulse response of each filter oscillates at a rate that is determined by the best frequency of the filter (Fig 2D), the average temporal width ($\Delta t$) of each filter is the same. Thus, unlike the cochlear filters, which have spectro-temporal resolution that varies with the filter best frequency, the Gabor filters of the Fourier representation have a constant spectro-temporal resolution.

Examining the relationship between spectral bandwidth and temporal width illustrates a key difference between Fourier and cochlear filters (Fig 2E). Theoretically, the uncertainty principle requires that the time-frequency resolution product of each filter satisfy the uncertainty principle [38]

$$\Delta t \cdot \Delta f \geq 1/\pi,$$

and equality holds for the Gabor filter case. Here we use three Gabor filterbanks with bandwidths of $\Delta f$ = 30 Hz, 120 Hz, and 480 Hz and corresponding temporal resolutions of $\Delta t$ = 10.6 ms, $\Delta t$ = 2.7 ms, and $\Delta t$ = 663 μs, respectively. All of the individual filters for each of the three Fourier filterbanks have identical time and frequency resolution, regardless of the best frequency. Thus, each filterbank is represented by a single point (Fig 2E). In contrast, the cochlear filters, have frequency-dependent bandwidths so that the time and frequency resolution of each individual filter depends on the filter best frequency and slightly exceeds the uncertainty principle theoretical limit (Fig 2E).

Although spectrographic representations are often treated as roughly equivalent, the differences in time-frequency resolution between the Fourier and cochlear filterbanks emphasize distinct sound features, dramatically impacting the spectrographic representation of speech, animal vocalizations, and other natural sounds (Fig 3). Narrowband Fourier spectrograms ($\Delta f$ = 30 Hz), for instance, tend to have detailed spectral resolution at the expense of limited

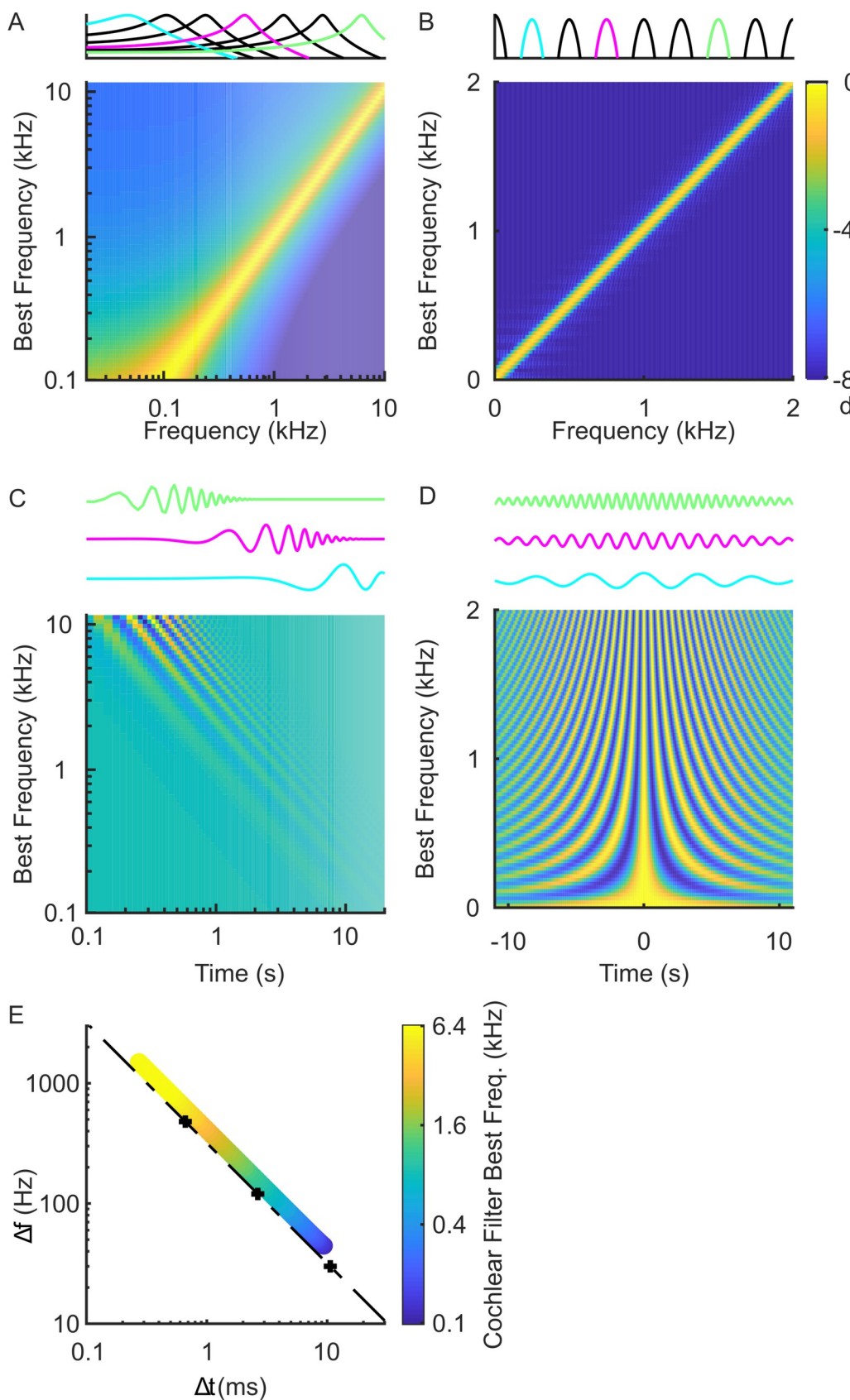

**Fig 2. Comparing Fourier and cochlear model filterbanks.** (A) Cochlear filter transfer functions are shown for model filters with best frequency between 0.1–10 kHz (color designates gain in dB). The cochlear filters are logarithmically spaced and have bandwidths that scale with frequency (proportional resolution). They exhibit a sharp high-frequency transition and gradual low frequency transition as observed physiologically for auditory nerve fibers. A subset of the transfer functions is line plotted above. Three selected filters (103.5, 830.0, 6653.5Hz) are shown in different colors and their corresponding time domain impulse responses are shown below. (B) The Fourier filterbank, by comparison, has constant resolution filters (30 Hz bandwidth shown here) that are ordered on a linear scale (shown up to 2kHz for clarity, and part of them are line plotted above, three examples are: 250, 750, 1500Hz). In the time domain, the cochlear filter impulse responses (C) have frequency dependent peak amplitudes and delays and the impulse response durations scale inversely with frequency. For visualization purposes and to allow for ease of comparison the impulse response line plots for the three examples are normalized to a constant peak amplitude (C, top). The Fourier filterbank filters, by comparison, have constant duration and are designed for zero delay (D). (E) shows the time ($\Delta t$) and frequency ($\Delta f$) resolution of the cochlear (colored circles) and three distinct Fourier filterbanks (+ symbols show $\Delta f$ = 30 Hz, 120 Hz, and 480 Hz). The dotted line represents the uncertainty principle boundary. Although the Fourier filterbanks are represented by a single point and fall on the uncertainty principle boundary, the time-frequency resolution of the cochlear filters is frequency dependent (colored circles).

temporal resolution while broadband Fourier spectrograms ($\Delta f$ = 480 Hz) have substantially faster temporal fluctuations and coarser spectral details (Fig 3). In speech, for instance, harmonic structure is evident in the narrowband Fourier spectrogram during voiced segments extending out to approximately 5 kHz (male talker; fundamental varies between ~100–170 Hz). However, the narrow bandwidth associated with these filters limits the filter temporal resolution and hence the fastest temporal modulations that can be resolved by this representation ($\Delta t$ = 10.6 ms, ~50 Hz upper limit). A broadband spectrogram, with coarser spectral ($\Delta f$ = 480 Hz) and higher temporal ($\Delta t$ = 663 µs) resolution, cannot resolve individual harmonics and, instead, exhibits periodic fluctuations at the fundamental frequency of voicing (Fig 3; vertical striations).

By using filters with frequency-dependent resolution, the cochleogram model emphasizes distinct spectro-temporal features in natural sounds. The cochleogram of speech, for instance, has approximately four resolved harmonics due to the relatively narrow filters for low frequencies as seen in the red (0.1–0.4 kHz) and magenta (0.4–1.6) panels highlighting a segment of speech (Fig 3). The broader higher frequency cochlear filters (black; 1.6–6.4 kHz), by comparison, are unable to resolve voicing harmonics, and instead generate detailed temporal modulations extending out to several hundred hertz (vertical striations visible upon zooming in; visible in the black and magenta panel). The high frequency filters also highlight formant structure which show up as coarse fluctuation in power across frequency (visible in the black and magenta panels). The cochleogram thus accentuates voicing harmonic structure in the low frequency channels while simultaneously producing voicing periodicity through the relatively broad high frequency cochlear filters. Similar distinctions are observed for nonharmonic sounds. For instance, the crackling fire has pronounced and transient modulation resulting from crackling embers (broadband pops between ~1–10 kHz) that is visible in the cochleogram and less pronounced in the narrowband spectrogram. Importantly, for all examples, Fourier spectrogram power is biased towards low frequencies, while the cochleograms have more evenly distributed power across channels.

## Cochlear filter tuning characteristics whiten the power spectrum statistics of natural sounds

Given the differences between the cochlear and Fourier representations, we next computed the spectrum statistics of natural sounds for both model representations and used entropy measures to evaluate the effectiveness of each decomposition. Specifically, we explore the hypothesis that the cochlear filters enhance the representation of natural sounds by "whitening" or flattening the output power spectrum, thus producing a more efficient neural representation.

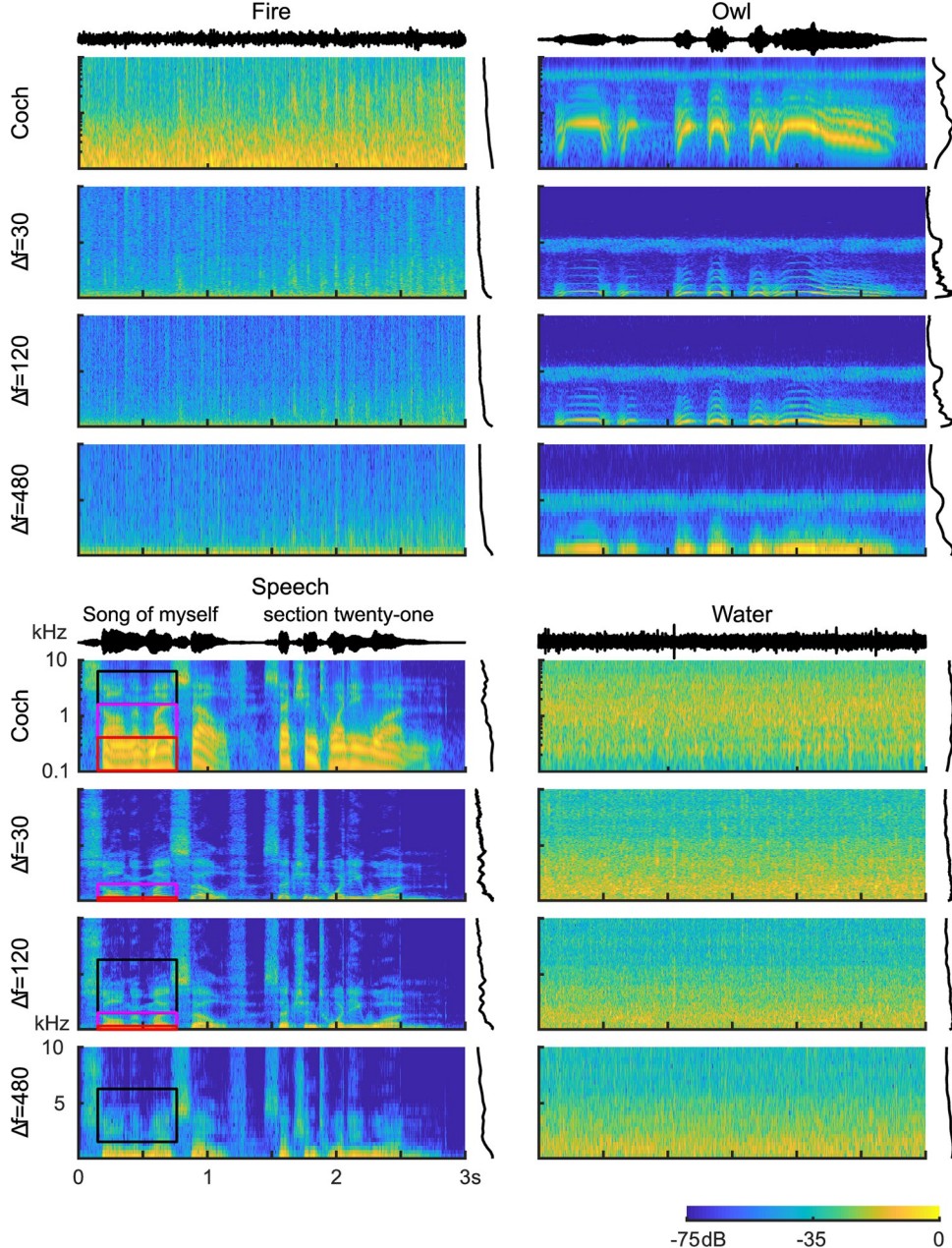

**Fig 3.** Example Fourier and cochlear model spectrogram decompositions for vocalizations and background environmental sounds: (A) Crackling fire, (B) owl vocalization, (C) speech, and (D) running water. Fourier-based spectrograms are shown for three different frequency resolutions ($\Delta f$ = 30, 120 and 480 Hz). The Fourier spectrograms tend to have higher power and details that are more concentrated at low frequencies, while the cochlear spectrograms have spectro-temporal components and power distributions that are more evenly distributed across frequency. Black (1.6–6.4 kHz), magenta (0.4–1.6 kHz) and red (0.1–0.4 kHz) boxes for speech (C) illustrate a regions of the Fourier or cochlear spectrograms that emphasize the voicing hormonic structure, second formant, and voicing temporal periodicity, respectively.

Theoretically, high spectral entropy is achieved whenever the power spectrum of a sound exhibits a uniform or a flat power distribution. From an encoding perspective, high entropy is thus achieved whenever the filters have outputs with uniform power so that the original signal power is spread equally across all filters (e.g., hair cells or neurons). For the Fourier

spectrographic decompositions, which have equal bandwidth filters, we expect that the highest entropy will be observed for white noise. By comparison, for the cochlear spectrographic model which has bandwidths that scale with frequency it is expected that the decomposition potentially boosts the output power at high frequencies. Since the high frequency filters integrate across broader bandwidths, the cochlear filters act to "whiten" the output for sounds with power spectra that decrease with frequency.

To characterize how the cochlear and Fourier filterbanks impact the filterbank spectrum output statistics for natural sounds, we first computed the Fourier- (Fig 4A) and cochlear-based (Fig 4B) power spectra for each sound category. Fourier-based power spectra tend to drop off with increasing frequency for all of the environmental sound categories tested. With the exemption of the Tamarin (slope = +1.9 dB/kHz, $\Delta f$ = 30Hz; Results for 120 Hz and 480 Hz shown in S1 Fig), vocalization sounds also exhibit a decreasing power trend with increasing frequency, while a white noise control sound has a flat Fourier spectrum (slope = 0 dB / kHz;

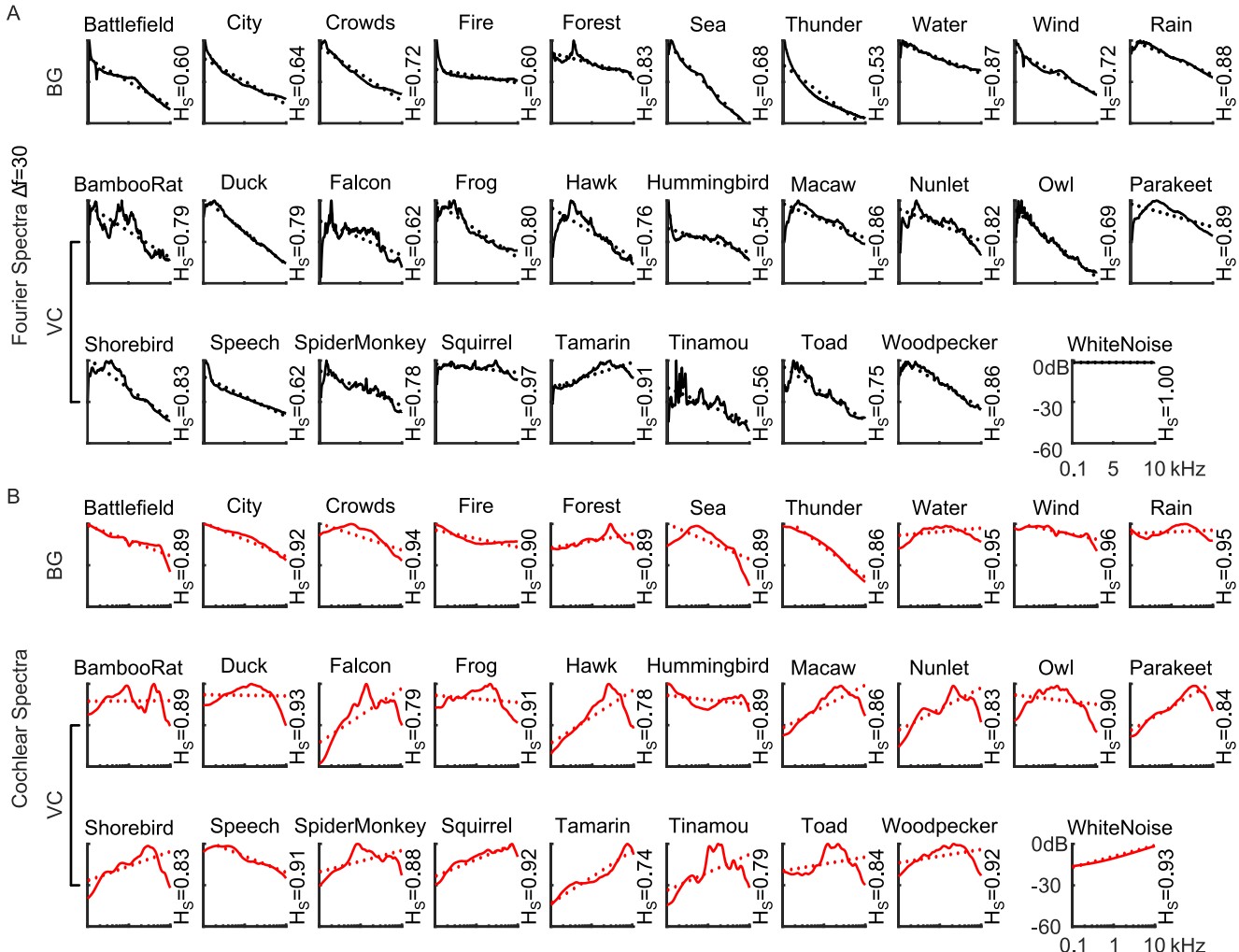

**Fig 4.** Spectra of vocalizations (VC) and background (BG) natural sound ensembles. Power spectra are shown for both the (A) cochlear and (B) Fourier-based model representations. Dotted lines represent the best linear fit between 0.1–10 kHz. All but one of the natural sounds have a Fourier spectrum with negative slope, while cochlear spectrums, by comparison, have more varied slopes (positive and negative) indicating a more even distribution of power across frequencies. The spectral entropy of each sound category is listed on the right side of the panel.

Fig 4A). The cochlear model power spectra, on the other hand, are generally more varied. Some vocalization categories, such as the hawk, tamarin and parakeets, tend to have cochlear spectra that increases with frequency; other categories, such as the bamboo rat and hummingbird sounds, have spectra that are relatively flat on average and, yet, other categories such as speech have somewhat decreasing power trends. Background sounds by comparison tend to be biased towards having decreasing power trends (e.g., city, thunder, and ocean sounds) or are relatively flat (e.g., rain, forest, and fire sounds). Without the bandwidth scaling, the observed flattening is not present in the cochlear representation and the results resemble the Fourier spectra (S5 and S6 Figs). This suggest that bandwidth scaling is a main factor for whitening the cochlear model representation.

We then compare the distribution of measured slopes for both filterbanks. To account for the fact that the two filterbanks have distinct frequency axes and the power spectrum slopes have different units (dB/kHz for Fourier; dB/octave for cochlear model), we normalized the slopes of each filterbank by their standard deviation (normalized by the standard deviation of the ensemble distribution), so that both filterbanks have slope distributions with SD = 1. For the Fourier decomposition, most vocalizations and background sounds have similar negative slope (standardized slopes = -1.8 vs. -1.9 standard deviations, vocalizations vs. background, respectively; $\Delta f$ = 30Hz; t-test, p>0.7). In contrast, the cochlear model standardized slopes tend to be smaller in magnitude, spanning both negative and positive values, and with an average slope that was not significantly different from zero (t-test, p>0.29). Interestingly, vocalizations are biased towards positive slopes (0.75±0.22, mean±SE; t-test, p<0.01) and backgrounds biased towards negative values (-0.41±0.25, mean±SE; t-test, p<0.01).

We next computed the spectral entropy of each sound for the Fourier and cochlear filterbanks as a way of assessing their encoding effectiveness. As a reference, the spectral entropy of white noise is highest for the Fourier filterbank (1.00 vs. 0.93 bits, Fourier vs. Cochlear; * on Fig 5B). This is consistent with the notion that white noise generates a flat power spectrum for the Fourier filterbank and, thus, ultimately is most efficiently represented with a Fourier like decomposition. When comparing all sounds, the measured spectral entropy of the majority natural sounds categories was larger for the cochlear over the Fourier decomposition (25 of 29 categories; Fig 5B), indicating that cochlear model filters produce a more efficient spectral representation. Across all natural sounds the cochlear model entropy (0.88±0.06, Mean±SD) is significantly higher than the Fourier based entropy (0.75±0.12, Mean±SD; $\Delta f$ = 30 Hz) regardless of the filter bandwidths used (paired t-test with Bonferoni correction, p<0.05; $\Delta f$ = 30, 120 or 480 Hz). When comparing vocalization and background sound categories (Fig 5C), we find that measured entropies for the background sound categories are higher for the cochlear decomposition (0.92±0.03 for cochlear; 0.71±0.12 for $\Delta f$ = 30; 0.59±0.18 for $\Delta f$ = 120; 0.52 ±0.19 for $\Delta f$ = 480; Mean±SD, t-test with Bonferoni correction, p<0.05). Similarly, the vocalization entropy for the cochlear filters was also higher than the Fourier filters (0.86±0.06 for cochlear; 0.77±0.12 for $\Delta f$ = 30, 0.71±0.14 for $\Delta f$ = 120, 0.65±0.15 for $\Delta f$ = 480; t-test with Bonferoni correction, p<0.05).

These comparisons demonstrate how cochlear model filters produce flatter spectra for both vocalizations and background sound categories. These findings are consistent with the hypothesis that cochlear filter decomposition whitens the cochlear spectrum of natural sounds ultimately producing a more efficient population representation [23,25]. Here we further propose that the output whitening is a direct result of bandwidth scaling for cochlear filters. To illustrate this effect, we show how the cochlear spectrum of natural sounds can be predicted directly from the Fourier spectrum by taking into account the residual accumulated output power that arises from cochlear bandwidth scaling. That is, since cochlear filters scale with frequency we propose that integrating the sound power spectrum across increasing bandwidths

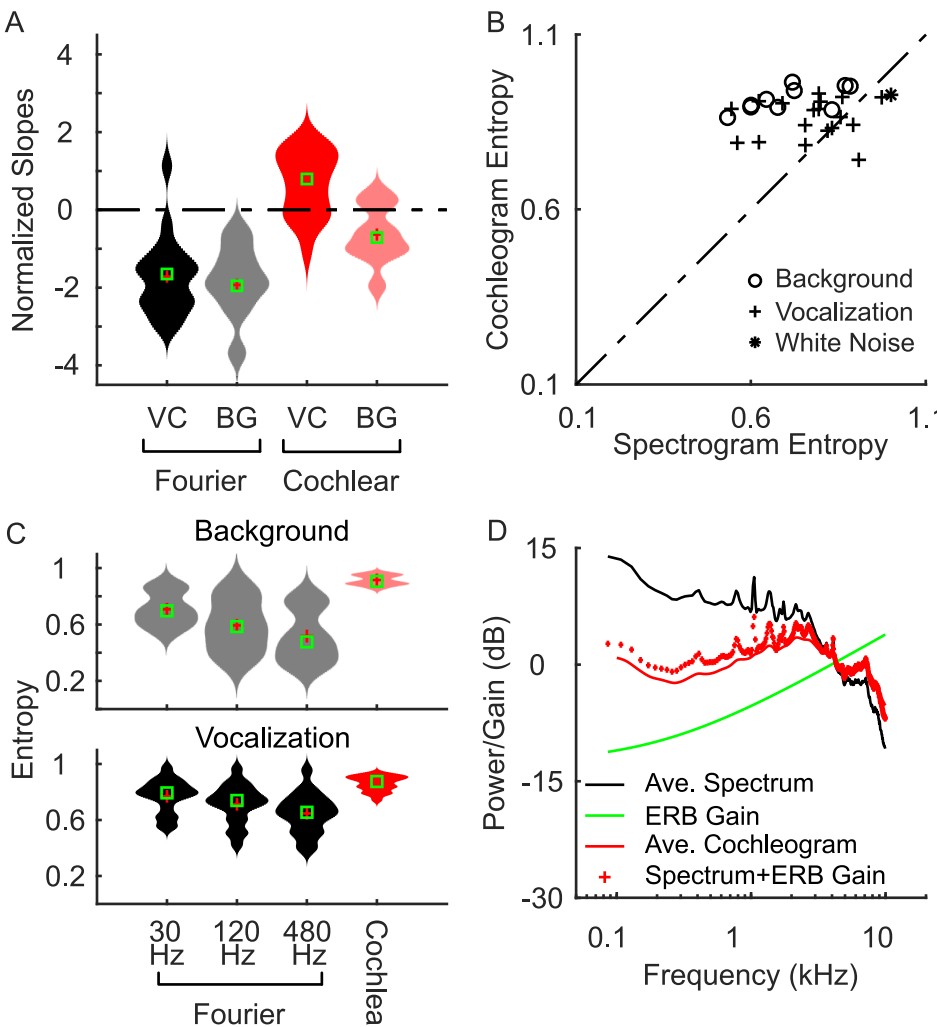

**Fig 5. Cochlear model bandwidth scaling whitens the power spectrum of natural sounds and maximizes spectral entropy.** (A) Violin plots showing the distribution of normalized slopes of the best regression fits to both the Fourier and cochlear models (from Fig 4). For both vocalization and background sounds, normalized spectral slopes for the Fourier decomposition are negative and not significantly different (t-test, p = 0.58). By comparison, vocalizations have positive and negative slopes for vocalizations and background sounds, respectively, with an average slope near zero (0.2) indicating a whitened average spectrum. (B and C) The cochlear model entropy is higher than Fourier-based entropy regardless of the Fourier filter resolution used (30, 120 or 480 Hz). (D) Bandwidth scaling predicts the cochlear filter whitening. The average Fourier power spectrum has a decreasing trend (black) whereas the cochlear power spectrum is substantially flatter (red, continuous). The gain provided by the cochlear filter bandwidths (green curve) increases and counteracts the decreasing power trend of the Fourier power spectrum. The cochlear power spectrum is accurately predicted by considering the bandwidth dependent gain (dotted red lines; bandwidth gain + Fourier power spectrum).

(with increasing frequency) allows the high frequency filters to accrue more power, which imposes a bandwidth dependent gain on the cochlear outputs (Fig 5, green). Fig 5D shows that the Fourier based power spectrum (average across all natural sounds) has a decreasing power trend (black curve) with increasing frequency while the average cochlear model power spectrum of natural sounds is substantially flatter (red curve). By imposing the proposed bandwidth dependent gain of the cochlear filters, we can accurately predict the cochlear power spectrum (Fig 5D). As seen, there is a strong correspondence between the actual cochlear

power spectrum (continuous red) and the predicted cochlear power spectrum (dotted red) with an average error of 0.75 dB. Thus, the cochlear model output power spectrum can be predicted by considering the Fourier power spectrum and adding the frequency dependent gain of the cochlear filters (in units of dB).

## The consequences of midlevel auditory filter tuning on the modulation statistics of natural sounds

Following the cochlear decomposition of sounds into frequency components, midbrain auditory structures such as the inferior colliculus carry out a second-order decomposition of sounds into spectro-temporal modulation components. Spectro-temporal modulations are critical acoustic features that strongly influence the perception and recognition of natural sounds. Here, by comparing Fourier-based, cochlear, and auditory midbrain representations, we explore the consequences of this secondary decomposition and propose that, by building on the cochlear representation, the tuning characteristics of midbrain auditory filters further enhance the representation of natural sounds.

Like the cochlear filters, auditory midbrain filters exhibit bandwidth scaling [8]. To evaluate how the midbrain auditory filters impact the representation of natural sounds, we compare the modulation statistics of the natural sound ensembles with Fourier, cochlear, and an auditory midbrain model representation. Here the Fourier spectrogram is passed through a set of modulation decomposition filters [2] and the outputs are used to compute the Fourier modulation power spectrum (MPSf). In the modulation domain (Fig 6C), the Fourier modulation decomposition filters have a constant modulation bandwidth (both spectral and temporal) regardless of the spectral or temporal modulation frequency being analyzed. In the spectrogram domain (Fig 6D), these modulation filters consist of spectro-temporal Gabor functions with constant duration and spectral resolution. Next, to characterize the modulation statistics obtained with a cochlear filter decomposition, we estimated the modulation power spectrum of the cochleogram (MPSc) [8]. Here, the cochleogram of each sound is processed through Fourier based set of modulation filters with equal modulation resolution, analogous to the MPSf (as in Fig 6C). Finally, we consider a midbrain-based representation by taking the cochleogram outputs and processing them through a modulation filterbank model of the auditory midbrain (Fig 6A). Here, unlike the Fourier based modulation filters used for the MPSf and MPSc, which have constant modulation resolution, the spectral and temporal modulation filter bandwidths are chosen to scale proportional to the best spectral and temporal modulation frequency of each filter, respectively (see Methods; Fig 6A). These modulation filters have a quality factor of 1, i.e., the spectral and temporal modulation bandwidths are equal to the best temporal and spectral modulation frequency, respectively, mimicking physiological measurements [8]. In the cochleogram domain (Fig 6B), these midbrain-inspired modulation filters resemble spectro-temporal receptive fields (STRFs) that account for the spectro-temporal selectivity of auditory midbrain neurons [36]. Like their neural counterparts, the model filters have durations that become progressively shorter for high modulation frequencies and have narrower tuning for high spectral modulation frequency filters. In other words, these midbrain model filter scale with modulation frequency and the filter durations and bandwidths are inversely related to the best temporal and spectral modulation frequencies, respectively.

Just as different spectrographic decompositions emphasize different sound features, these three modulation decompositions emphasize distinct modulation features. For example, the narrowband Fourier MPS ($\Delta f$ = 30 Hz) emphasizes spectral over temporal modulation features, since the temporal modulations of all sounds here tend to be <50 Hz (Fig 7A, $\Delta f$ = 30 Hz; results for $\Delta f$ = 120 and 480 Hz are shown in S2 Fig), and the spectral modulations have

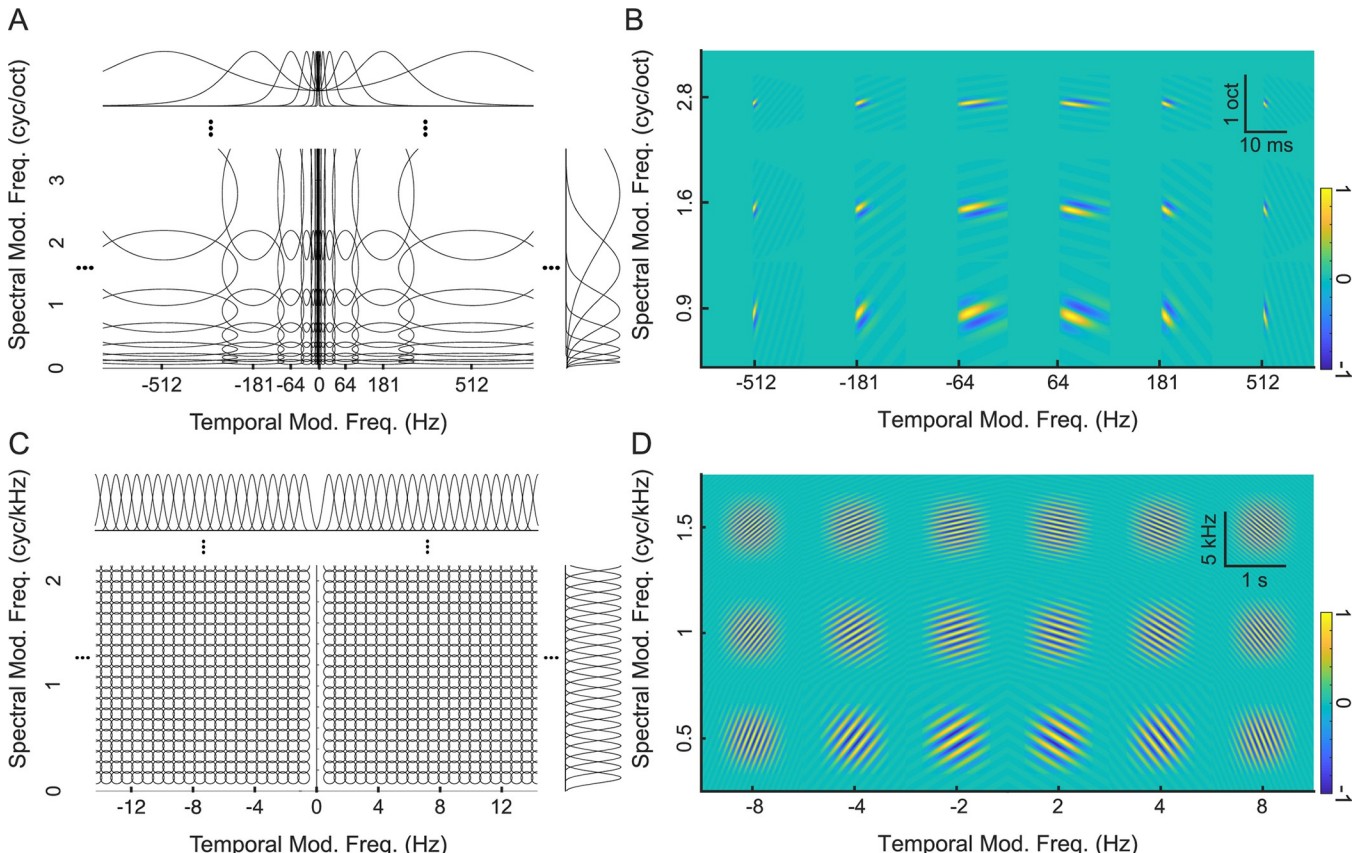

**Fig 6. Fourier and midbrain modulation filterbanks.** Modulation decomposition filters are shown for (A) the midbrain filterbank and (C) the Fourier-based filterbank, with each transfer function contoured at the 3dB level (50% power). Note that the Fourier-based modulation filters have equal resolution in both spectral and temporal dimensions, whereas the midbrain modulation filters have proportional resolution as observed physiologically (i.e., bandwidth scaling). The corresponding STRFs are shown for both the (B) midbrain filterbank and (D) Fourier-based filterbank. Note that the Fourier-based STRFs have equal duration and bandwidth whereas the durations and bandwidths scale for midbrain filters.

units of cycles/kHz (Fig 3). This filter structure ultimately emphasizes detailed spectral fluctuations with an upper limit in the range of ~15 cycles/kHz as determined by the 90% energy contours of all sounds, including white noise (black contours in Fig 7A). The equal resolution spacing of the spectral modulation filters also emphasizes harmonically related components, such as the mode between 5–10 cycles/kHz is created by harmonics in voiced speech [1].

The cochleogram MPS includes substantially higher temporal modulations, but at the expense of having substantially lower spectral resolution for the cochlear filters, which on average are broader than those of the narrowband Fourier spectrogram. The 90% power contours in the MPSc for white noise extend to 500 Hz (black contours in Fig 7B), well beyond the narrowband MPSf (limited to ~50 Hz). Across sound categories, the range of temporal modulations in the MPSc was highly variable. For example, vocalizations have 90% power contours that extend beyond 50 Hz at zero spectral modulation (249.6±137.5 Hz, mean±SD) and these were substantially higher than the corresponding contours for the narrowband MPSf (39.0 ±9.9, $\Delta f$ = 30 Hz; mean±SD). In the spectral modulation dimension, the natural sounds are largely limited to less than 4 cycles/octave. Thus, cochlear filters appear to accentuate temporal features at the expense of spectral modulation content.

While the cochlear and Fourier-based MPS accentuate unique set temporal and/or spectral modulation features, the midbrain auditory representation further transforms the modulation

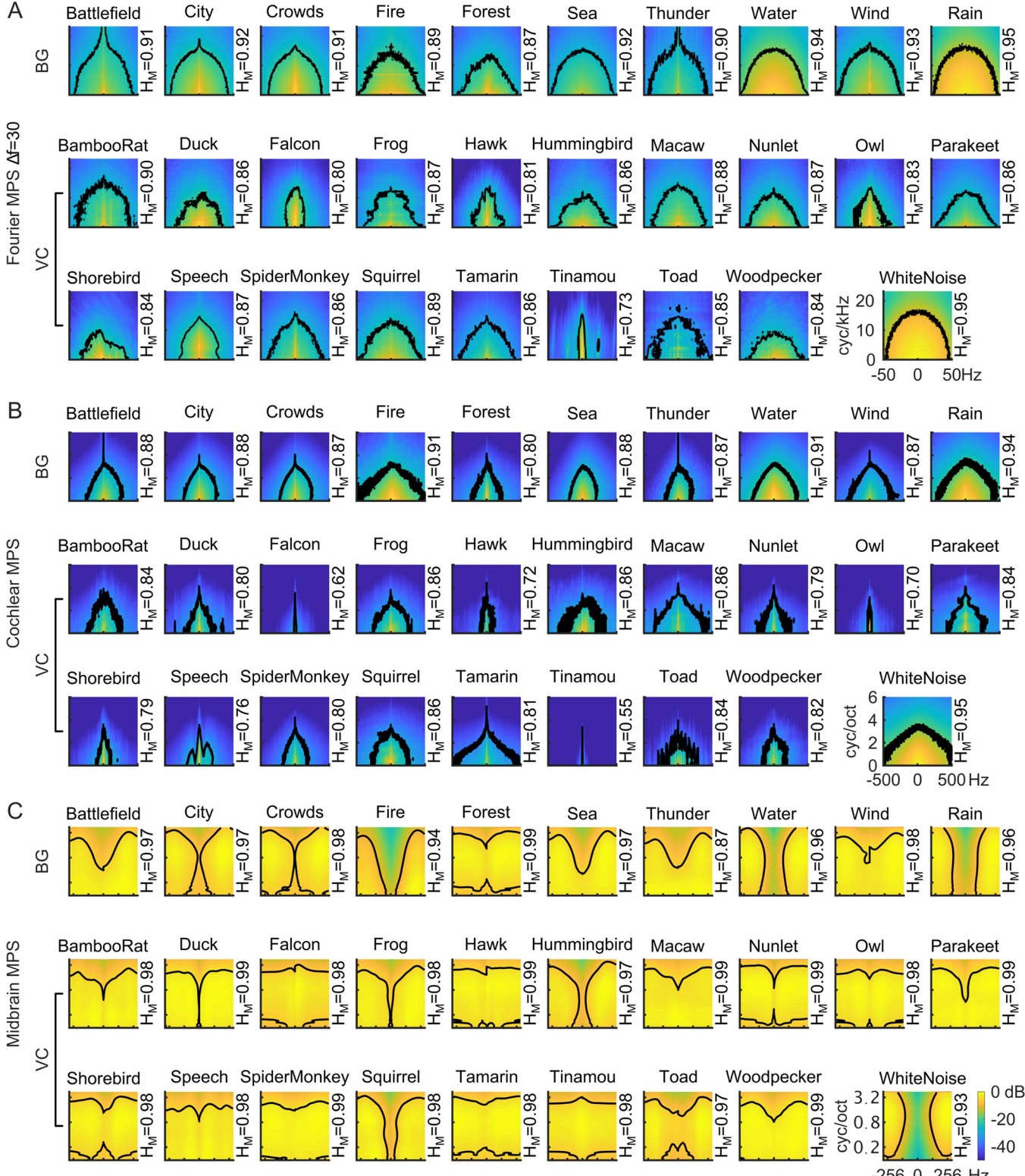

**Fig 7.** Modulation power spectra of natural sound ensembles including vocalizations (VC) and background sounds (BG). The modulation power spectrum is shown for the (A) Fourier-based decomposition (Δ*f* = 30 Hz), (B) cochlear model decomposition and (C) midbrain model decomposition. Whereas the Fourier MPS and cochlear model MPS overemphasize low frequency spectral and temporal modulations, the midbrain model MPS is substantially flatter. Black contours in each graph designate the MPS region accounting for 90% of the total sound power. The modulation entropy of each sound category is listed on the right side of the panel.

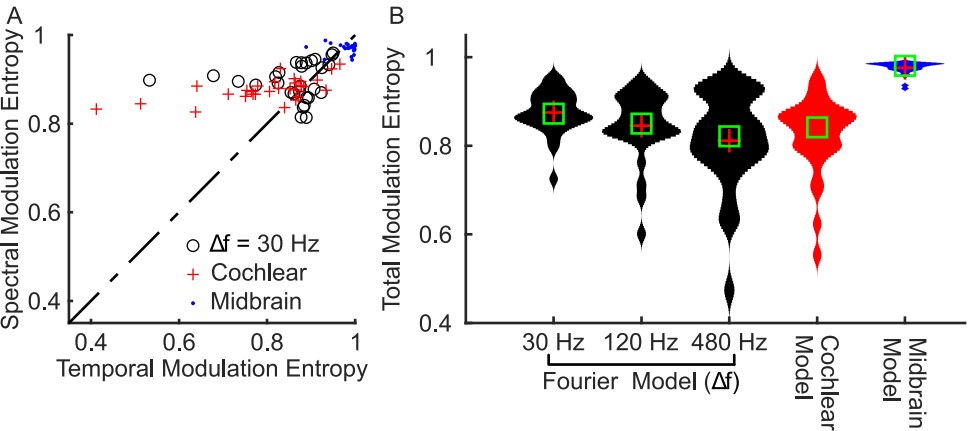

**Fig 8. Midbrain model decomposition maximizes the modulation entropy of natural sounds.** (A) Spectral and temporal modulation entropy are significantly higher for the midbrain model when compared against Fourier (black; $\Delta f$ = 30 Hz) and cochlear model (red). (B) The total modulation entropy is highest for the midbrain model when compared against Fourier and cochlear models.

content. While the power tends to drop off with increasing temporal and spectral modulation frequency for the MPSf and MPSc, the modulation statistics derived through the midbrain model are far more uniform (Fig 7C; also shown using individualized color scale in S3 Fig). Here the midbrain MPS of all natural sounds is substantially flatter than either the MPSf and MPSc across both spectral and temporal modulation dimensions when all natural sounds are considered, yet, the MPSm of each sound ensemble is still unique and discernible.

To assess the efficiency of each of the three modulation representations, we measure modulation entropy. As with the spectrographic representations, the bandwidth scaling of the midbrain MPS leads to increased entropy compared to the Fourier and cochlear MPS (Fig 8). This pattern occurs for both the spectral and temporal modulation entropy alone (Fig 8A), as well as the total modulation entropy for the natural sounds (MPSf = 0.88±0.05, $\Delta f$ = 30 Hz; MPSc = 0.82±0.09; MPSm = 0.98±0.01; mean±STD; Fig 8B), even though the modulation entropy for white noise was comparable for the three representations (MPSf = 0.95; MPSc = 0.95; MPSm = 0.93). The dramatic differences in modulation entropy are not simply the consequence of the different range of modulations used for the modulation entropy calculation (see METHODS), but rather, reflect the statistical structure of the natural sounds. Collectively, these findings suggest that midbrain modulation decomposition produces a "whitened" representation of natural sound modulations that reduces redundancy by more equitably activating all elements of the modulation filterbank.

## Modulation filter bandwidth scaling as a mechanism for whitening the spectro-temporal modulation content of natural sounds

As for cochlear filters, where bandwidth scaling serves to whiten the neural representation natural sounds, modulation filter bandwidths derived perceptually [26,27] and physiologically in auditory midbrain [8] scale with the modulation frequency of sounds. Here we test and propose that bandwidth scaling for midbrain auditory filters provide a boosting mechanism for equalizing the midbrain MPS of natural sounds. Fig 9A illustrates that the for the average natural sound MPSc, power drops off with increasing spectral and temporal modulation frequencies, whereas the MPSm is substantially flatter (Fig 9B). We propose that by integrating across broader modulation bandwidths (with increasing spectral or temporal modulation frequency)

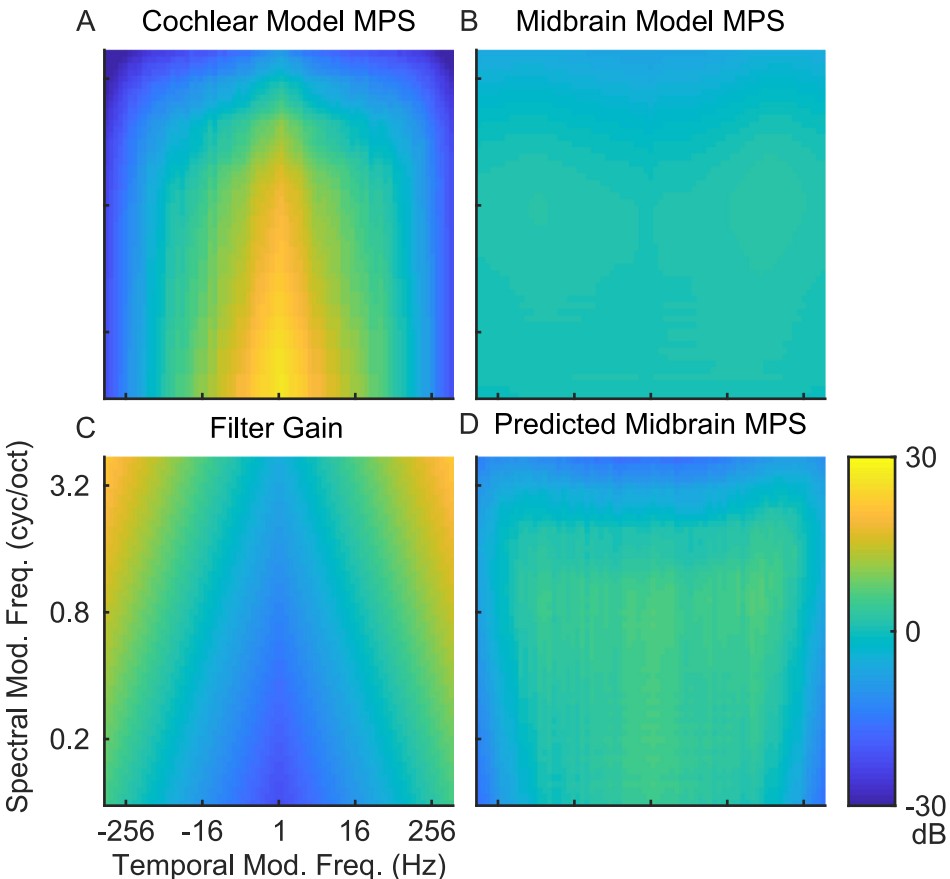

**Fig 9. Modulation bandwidth scaling in the midbrain model accounts for the modulation whitening.** Averaging over all sounds, the (A) cochlear model MPS overemphasizes low temporal and spectral modulations whereas the (B) midbrain model is substantially flatter MPS. (C) Residual gain of the midbrain modulation filters arising from bandwidth scaling. (D) The predicted midbrain MPS obtained by adding the Cochlear MPS (A) and bandwidth-dependent gain (C) accounts for the whitened output of the midbrain model.

midbrain filters impose a modulation frequency depend gain on their output (Fig 9C). Indeed, when we apply this gain to the MPSc it produces a substantially flatter output that matches the observed MPSm. Thus, the power gain that results from modulation bandwidth scaling naturally counteracts the decreasing power trend observed in the cochlear model MPS, thus providing a modulation frequency dependent gain mechanism that whitens the modulation spectrum outputs of natural sounds.

## Discussion

Here we examined how bandwidth scaling in the cochlea and midbrain influence the representation of natural sound spectra and modulations. Our findings are broadly consistent with the efficient coding hypothesis whereby sensory systems evolved to efficiently transduce and reduce redundancies in the statistical structure of natural sensory signals [18,42]. Both peripheral and mid-level auditory structures have scale-dependent and spectro-temporally compact filters, analogous to a multi-dimensional wavelet decomposition of sounds. These filters differ from conventional Fourier representations, which lack scaling and have constant spectro-temporal resolution. The peripheral and mid-level bandwidth scaling jointly equalizes the power

in the neural outputs in three dimensions (frequency, spectral modulation, and temporal modulation), which produces a more equitable and efficient representation of natural sounds. These whitening transformations may have implications for neural coding and perception, as well as for development of audio codecs, speech and sound recognition, and auditory prosthetics.

## Efficient representations of natural sounds

Although previous studies have shown that basis sets optimized for representing natural sounds can, in some cases, match the filter characteristics observed in the cochlea [23,25] and the auditory midbrain [24], here we directly examine consequences of filter characteristics known to exist physiologically. The main insight from our study is that bandwidth scaling in the cochlea and auditory midbrain provides a mechanism for hierarchically whitening the second- (power spectrum) and fourth-order (modulation spectrum) statistics of natural sounds. For both the spectrum and modulation spectrum of natural sounds, sound power decreases systematically with increasing frequency (or modulation frequency) and both cochlear and midbrain filter bandwidths scale to counteract this dependency. Having larger bandwidths at high frequencies allows neurons to integrate over a larger extent of frequencies and thus accumulate more of the weak high frequency signals. This in turn produces a boost in the output power for these weak high frequency signals at the expense of having coarser spectral (cochlear filters) or modulation (midbrain filters) resolution.

At the cochlear level, our results indicate that bandwidth scaling is matched to the power spectrum statistics of environmental sounds and vocalizations. This is consistent with previous work from Lewicki [23,25] showing that the optimal filters for representing natural sounds are dependent on the stimulus categories used during training and that compact filters resembling those in the cochlea are obtained only when both vocalizations and environmental sounds are included. In our case, individual sound ensembles have cochlear spectra that are quite varied and on their own are not fully whitened. When considering only vocalizations, the cochlear model outputs overemphasized high frequencies, producing positive cochlear spectrum slopes and indicating that the bandwidth scaling overcompensates for the decreasing power spectrum trend in the sounds. By comparison, for background sounds, the cochlear model outputs overemphasized low frequencies, producing negative slopes and, consequently, lower entropies (Fig 5A and 5C). Thus, the outputs for vocalization or environmental sounds ensembles individually produced a biased, suboptimal output representation although for both cases they are closer to a whitened output spectrum when compared against the Fourier representation. Despite these individual ensemble biases, vocalizations and environmental sounds counterbalanced each other and produce combined cochlear spectrum that is, on average, whiter than either category.

Despite the observed whitening, a reduction of power of ~10 dB is still observed at the highest frequencies for the average cochlear spectrum (Fig 5D; red). One additional mechanism not explicitly accounted by our cochlear model that could further whiten the cochlear spectrum is the fact that the distribution of hair cells with different best frequencies varies along the cochlear spiral. Our model assumes that frequencies follow octave spacing, yet the cochlear spiral exhibits a nonlinear frequency versus position function spanning 10 octaves for human hearing (20 Hz– 20 kHz) that deviates from an octave approximation at low frequencies [43]. Using Greenwood's model of the human cochlea [31,43] and the fact that there are roughly 100 hair cells per mm [44] we estimated ~80 hair cells for the lowest octave of hearing (20–40 Hz) and ~500 hair cells over the last octave (10–20 kHz). Under the assumption that sound power is integrated by the auditory system across peripheral receptors, this would correspond

to an increase in output power of ~ 8dB at the high frequencies (S4 Fig). Thus, although natural sounds are generally biased towards low frequencies (Figs 4 and S5) and many auditory phenomena dominate the low frequency range of hearing [45], this frequency dependent boost in the integrated cochlear power may further whiten the cochlear representation of natural sounds, thus extending the overall range of hearing.

Following power spectrum whitening at the cochlear stage, the modulation filterbank stage further whitens the modulation representation of natural sounds. Just as with the cochlear filters, bandwidth scaling in this mid-level auditory model appears to be critical for this secondary form of whitening. Neurons in the auditory midbrain have quality factors of ~1 such that modulation bandwidths scale proportional to modulation frequency [8]. Incorporating this simple observation in our model produces a bandwidth-dependent gain that precisely counteracts the 1/f modulation spectrum statistics of natural sounds.

The choice of spectro-temporal representation impacts the interpretation and modeling of neural data. Although spectro-temporal receptive fields (STRFs) are widely used to study peripheral and central auditory coding, findings differ depending on whether sounds are represented using synthesis envelopes, spectrograms, or cochlear model representations [6,46–49]. A recent study demonstrated that using a cochlear based model representation to derive cortical STRFs provides higher predictive power over other spectro-temporal representations [49]. This suggests that filters with physiologically-based spectrographic representations better capture important spectro-temporal features that are encoded at the cortical level.

Our approach differs from various prior studies which have derived optimal basis sets for representing natural sensory stimuli and testing the efficient coding hypothesis [23,24,50,51]. Although these studies employ a rigorous framework to test a computational theory, they nonetheless require assumptions about the nature of the proposed code and the optimization strategy used. For example, these models often assume linear basis sets which don't account for nonlinear characteristics of neural processing and often employ objective functions that are not biologically driven. More recent studies have overcome such limitations by employing deep neural network and behaviorally guided objective functions for optimization, which are presumably more biologically relevant [52,53]. Nonetheless, such models often have tens of thousands of parameters and can be difficult to interpret mechanistically. In our case, rather than optimizing a model, we employed a model with known biological constraints to develop a mechanistic explanation of the acoustic representation. This allowed us to demonstrate how sound whitening is achieved for multiple auditory features across multiple levels of auditory processing. In future studies, it would be valuable to derive a jointly optimal multi-stage filterbank in order to further identify optimal strategies and mechanisms for natural sound processing.

In addition, the observed whitening is likely mechanistically different from whitening in other sensory modalities and may be unique to audition. For instance, although multiple levels of whitening are observed in the visual system the known mechanisms differ from those described here. Whitening of visual scenes in the lateral geniculate nucleus is achieved by temporal decorrelation of the spike trains that occurs at the individual neuron level and which is restricted to low frequencies (<15 Hz) [54]. In primary visual cortex, additional whitening is achieved through nonlinear interactions of the classical and nonclassical receptive fields of individual neurons, which again are restricted to low frequency information (<36 Hz) [55]. In our case, whitening is an ensemble level phenomena that involves multiple tuned filters and which involves temporal information exceeding several hundredths of Hz.

Overall, our results demonstrate that whitening of multiple sound dimensions can be achieved hierarchically across multiple levels of auditory processing. Whitening in the cochlear model stage is restricted to sound spectra; whereas the mid-level stage whitens temporal and

spectral modulations. On the one hand, such a three-dimensional neural representation serves to equalize the statistics of natural sounds with well-known redundancy, such as the 1/f modulation power spectra [56–58] and varied and non-white spectro-temporal correlation statistics [59,60]. This whitening is achieved by having filters with variable resolution (either in frequency or modulation space). Neurons that integrate weak signals, such as fast temporal modulations, have broader bandwidths and thus integrate over a broader range of feature space, magnifying these weak signals and assuring that they are encoded and ultimately perceived. Although other forms of efficient coding due to adaptation, sparsity, or nonlinearities may coexist alongside these effects [21,61–63], here we focused on how bandwidth scaling distributes computational and metabolic resources evenly across a neural population, assuring that all neurons are utilized and contribute similarly to the neural representation.

## Implications for perception of natural sounds

The perception of acoustic attributes such as frequency, intensity, and modulation have been studied extensively over the past century; yet most perceptual studies do so without considering the neural transformations involved and their impact. Given that auditory filters emphasize a unique subset of acoustic features, we propose that they influence the perceived qualities of natural sounds and ultimately underlie perceptual abilities.

The auditory midbrain decomposes sounds into modulation components and several studies have proposed that its anatomical layout and receptive field characteristics could underlie several phenomena in audition. The laminar spacing and frequency bandwidths in auditory midbrain have been proposed to contribute to critical band perceptual resolution [64], and neural modulation bandwidths match those derived from perceptual measurements in humans [8,26,27]. Furthermore, decoding brain activity in auditory midbrain replicates perceptual trends for human texture perception [65]. Together, these results suggest that the mid-level auditory representation already contains spectro-temporal features that predict various aspects of natural sound perception.

Studies using physiologically-inspired representations of natural sounds also support the notion that peripheral and mid-level filtering transformations strongly shape the perception of natural sounds. For instance, water sounds exhibit scale-invariant power spectrum statistics and realistic acoustic impressions can be generated as a superposition of scale invariant gamma tone filters that mirror the cochlear filters [66]. Realistic synthetic impressions of "textures" sounds, such as crowd noise, wind, and running water can be generated with a generative model of the peripheral and mid-level auditory system; yet, removing the bandwidth scaling present in this model by using equal resolution filters, either in the peripheral or modulation filters, produces sound impressions that are less realistic [59]. The choice of representation also dramatically impacts word recognition accuracy for vocoded speech, since equal resolution filters tend to yield low recognition accuracy while filters optimized for efficient coding (with bandwidth scaling) substantially improve word recognition accuracy [67]. Collectively, these studies suggest that filterbank models that scale and mirror known physiology accentuate perceptually important features and thus generate more realistic and identifiable sound impressions.

Spectro-temporal modulations are also critical for speech perception, contributing to various perceptual attributes such as voice quality and pitch, vowel and consonant perception, phonetic and word segmentation and, ultimately, speech recognition or discrimination abilities. As demonstrated, different spectro-temporal decompositions accentuate a unique set of spectro-temporal features which produce distinctly different spectro-temporal outcomes. Thus, the unique differences in Fourier based versus cochlear representations can ultimately

lead to different interpretations of the cues that are important physiologically and perceptually. For instance, formants show up as relatively coarse fluctuations in power across frequency that are visible in both the Fourier based and cochlear representation. In both instances, these show up in modulation filters with low spectral modulation, yet they appear more compressed in the cochlear model as a result of octave spacing. Voicing pitch on the other hand, is even more dramatically impacted by the spectro-temporal representation. When speech is analyzed using narrowband equal resolution Fourier based filters, temporal modulations are severely limited (<50 Hz) while voicing harmonic content (spectral modulation) related to pitch is accentuated [1]. This harmonic content is a critical cue for voice quality and gender identification. In contrast to conventional spectrograms, the cochlear model only extracts a few harmonics for the low frequency range of hearing, yet it accentuate temporal information at the high frequencies. Such frequency dependent transformation is likely critical for perception and coding of speech and perceptual models need to consider such differences in the sound representation. There is a longstanding debate on whether the neural representation of pitch relies predominantly on temporal or spectral features of sounds (i.e., harmonicity versus periodicity) dating back to Helmholtz [68] and whether the neural representation itself is temporal or rate based in nature. Harmonic structure in sounds can be represented as a place-rate code implying a spectral analysis, which is particularly true for very low frequencies (<1000 Hz) where narrow cochlear tuning can resolve harmonic content. However, for higher frequencies cochlear outputs exhibit periodic temporal modulations if the harmonics are unresolved by the cochlear filters. There is also evidence that even nonharmonic periodic sounds (e.g., modulated noise) can produce weaker forms of pitch [14] and strongly drive periodic neural activity [37], indicating that harmonicity is likely not the sole determinant of pitch. Although spectral features are often regarded as dominant features in natural sounds, the auditory model analyzed here—and the physiological results the model is based on [8,28,69]—also implicates temporal structure as an important acoustic factor for representing natural sounds, speech, and pitch.

## Implications for audio coding and recognition systems

Audio and sound recognition technologies have dramatically improved over the past few decades. However, machine systems often perform poorly when recognizing sounds in complex environments with background noise, and cochlear implant and hearing aid technologies provide marginal benefits in noisy conditions. Here we suggest that these technologies could benefit from two physiologically-inspired sound processing strategies: 1) preserving detailed temporal information and 2) including bandwidth scaling. Previous work has shown that detailed temporal information is critical for human speech perception in noise [70], and bandwidth scaling in texture synthesis models yields more realistic impressions of natural sounds [59]. Although sound recognition systems often use the mel-spectrogram, which applies filters with spacing and bandwidths that scale and mirror human perception and physiology, these are applied to narrowband Fourier spectrograms that have limited temporal modulation content and fine structure and which are often limited to < 50 Hz. The cochlear filters used here, on the other hand, are applied to the sound waveform directly and preserve fine temporal modulations extending out to ~1000 Hz [35]. Here we have shown how bandwidth scaling in the cochlea and midbrain may act to hierarchically whiten natural sound representations, as well. Such physiologically inspired whitening of the acoustic space could potentially improve audio coding and lead to improvements in automatic speech recognition and prosthetic technologies, particularly for adverse and noisy conditions.

## Supporting information

**S1 Fig. Fourier power spectra for natural sounds with different resolutions.** Power spectra for all sound categories are analyzed using the Fourier-based model with resolutions: 30, 120 and 480Hz.
(PDF)

**S2 Fig. Fourier modulation power spectra of natural sounds with different resolutions.** Modulation power spectra for all sound categories are analyzed with the Fourier model with resolutions: 30, 120 and 480Hz.
(PDF)

**S3 Fig. Modulation power spectra of natural sounds for the auditory midbrain model.** Each sound category is plotted as in Fig 7, except that each is normalized to an individual power range and colorscale for visual clarity.
(PDF)

**S4 Fig. Predicted frequency dependent cochlear gain arising from hair cell density along the cochlear spiral.** (A) Frequency-position function for the human cochlea proposed by Greenwood [31,43] is broken up into 1 octave segments spanning low (20 Hz, blue) to high (20 kHz, red) frequencies. The lowest octave range (20–40 Hz) spans ~0.8 mm of the cochlear spiral while the highest octave spans ~5 mm. (B) Predicted hair cell count for different frequency ranges (1 octave segments) obtained by assuming 100 hair cells / mm [44]. Hair cell counts increase with increasing frequency resulting in ~5 times as many hair cells per octave for high frequencies. (C) Predicted cochlear output gain of our model for different 1 octave segments arising from hair cell density. The increased hair cell density per octave at high frequencies produces an ~8 dB increase in our model output power relative to the lowest frequencies.
(PDF)

**S5 Fig. Bandwidth normalized cochlear spectra.** The cochlear spectrum of natural sounds (outputs of the cochlear model) shown in Fig 4 (panels B) were normalized by the cochlear filter bandwidths. This provides the cochlear output power per Hz. The results for each natural sound closely resemble the Fourier spectrum of Fig 4A suggesting that flatting of the cochlear spectrum observed in Fig 4A arises because of the cochlear bandwidth scaling. Dotted lines correspond to the linear regression fits for each natural sound category.
(PDF)

**S6 Fig. Sloped distribution for the bandwidth normalized cochlear spectra.** The normalized slope distributions (shown as Violin plots) for vocalization and background sounds exhibit similar trends as for the Fourier power spectrum (compare with Fig 5A).
(PDF)

**S1 Text. Mathematical proofs, supporting figures and sound compilations.**
(DOCX)

**S1 Table. Sound list.** List of all sounds used for analysis, including their duration, categories, sources, etc.
(XLSX)

## Author Contributions

**Conceptualization:** Fengrong He, Ian H. Stevenson, Monty A. Escabí.

**Data curation:** Fengrong He, Monty A. Escabí.

**Formal analysis:** Fengrong He.

**Funding acquisition:** Ian H. Stevenson, Monty A. Escabí.

**Investigation:** Fengrong He.

**Methodology:** Fengrong He, Ian H. Stevenson, Monty A. Escabí.

**Project administration:** Monty A. Escabí.

**Resources:** Monty A. Escabí.

**Software:** Fengrong He, Monty A. Escabí.

**Supervision:** Monty A. Escabí.

**Validation:** Fengrong He, Ian H. Stevenson, Monty A. Escabí.

**Visualization:** Fengrong He, Ian H. Stevenson, Monty A. Escabí.

**Writing – original draft:** Fengrong He, Monty A. Escabí.

**Writing – review & editing:** Fengrong He, Ian H. Stevenson, Monty A. Escabí.

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
