## [Decision Letter · Decision Letter 0]

22 Jun 2022

Dear Dr. Escabi,

Thank you very much for submitting your manuscript "Two stages of bandwidth scaling drives efficient neural coding of natural sounds" for consideration at PLOS Computational Biology.

As with all papers reviewed by the journal, your manuscript was reviewed by members of the editorial board and by several independent reviewers. In light of the reviews (below this email), we would like to invite the resubmission of a significantly-revised version that takes into account the reviewers' comments.

We cannot make any decision about publication until we have seen the revised manuscript and your response to the reviewers' comments. Your revised manuscript is also likely to be sent to reviewers for further evaluation.

Sincerely,

Xuexin Wei

Associate Editor

PLOS Computational Biology

Samuel Gershman

Deputy Editor

PLOS Computational Biology

Reviewer's Responses to Questions

**Comments to the Authors:**

Reviewer #1: In “Two stages of bandwidth scaling drives efficient neural coding of natural sounds”, He, Stevenson and Escabi show that the double auditory filter bank performed by cochlea to auditory midbrain performs a whitening operation on natural sounds, yielding neural responses with maximum entropy and thus potentially maximum efficiency. To get to this conclusion, they use a physiologically realistic model of the auditory cochlea and of the modulation filter bank found in the auditory midbrain. The lab is very familiar with the second since they have acquired the neuro physiological data. Overall, the paper is well written, and the results are sound. On the one hand, I was particularly impressed by the clear description of the modulation filter bank and the modulation power spectrum of natural sounds. Just for that part, the paper will be very useful as a tutorial. On the other hand, I have made a list of shortcomings about the analysis and conclusions that need to be addressed – most of them relate to conclusions on optimal representations based on single neuron analyses instead of populations. There are also some important issues in terms of temporal structure that are muddled in the paper and need clarification.

Major Points:

1. I am not completely convinced by the whitening of the frequency spectrum by the cochlear filters. Yes – it is true that at the level of a hair cell or auditory nerve fiber that gain of the high-frequency channels is higher because the tuning bandwidth is larger. But at the level of the ensemble, there are also more bands (more neurons) in the low frequency range than the higher frequency range. That effect is not well represented in the plots of figure 4. Shouldn’t one consider the density of frequency bands as well? Similarly, the conclusions do not quite jibe with the critical bands obtained in loudness summation experiments. That body of work would suggest that we are performing a low pass filtering on the sounds. I believe you need to address these issues. At a very minimum, you can mention that you are considering optimal principles at a single neuron level and clarify the apparent contradictions that I have raised here.

2. Related to point 1, one can also imagine estimating ensemble entropy measures that are also take into account the correlations across frequency bands. The results and conclusions might be quite different.

3. I believe that the same comments (1 and 2) apply to the modulation filtering argument. Again, I agree that midbrain neurons with faster modulations have wider modulation bandwidth tuning (an empirical fact that you have shown !) and that this results in greater gain in that area of the MPS for a single neurons – but there are less of then with this tuning as well – right? (you have that data !).

4. I appreciate the focus on the cochlear representation, followed by the analysis of the midbrain representation. But the picture of auditory periphery processing is incomplete. Low-frequency auditory nerve fibers also phase lock to the actual waveform of the signal in the narrowband filtered signal, providing additional information that could further increase spectral resolution and the detection of voice pitch. The temporal structure you describe (e.g. in the discussion in lines 800-807) results for the high frequency cochlear filters and the corresponding fast amplitude modulations captured by your model. The correlated phase locked activity at low frequencies clearly also carries information on fast time structure. By the way this is the sTFS that is discussed in the paper that you cite in [61] in the context of your work that discusses fast temporal modulations. I think this could lead to further confusion on this temporal coding topic that is already poorly understood by many. It is possible that the sensitivity to phase for low frequency AN fibers end up contributing to the high temporal modulations sensitivity that you are describing but that is not part of your auditory model.

5. This is somewhere between a major and minor point. As you know, prior computational papers that have addressed optimal coding strategies (and verified Barlow’s hypothesis) have often started from an objective function (e.g. it would be entropy here) that they then maximize to find the best set of filters (as done by Lewicki for example using a sparsity objective function). Your approach is a bit different in that you just examine a biologically inspired model and show that it performs whitening. This is clearly interesting but maybe not quite as powerful. It might be too much to ask to try to find the optimal double filter bank but you should probably discuss this more clearly and think about how one might do this.

Minor Points:

1. The introduction does a very good job at summarizing what is known about natural sound statistics and auditory representations and also at introducing the modulation power spectrum. It falls a bit short when introducing the question addressed in the paper in the last paragraph.

2. 457 sound segments is probably more than sufficient but one will notice that it is much less than the typical number of images used by researchers investigating visual object representation.

3. L 463. Since you are talking about cochlear filters it is a bit weird to add at the end of this sentence “..analogous to cochlear filter tuning”. I know you mean model filters are analogous to the actual physiological filtering, but I bet that many readers will get stuck here.

4. In figure 2C and in the text l457-476, you might also want to discuss the gain of the filters for different frequencies. The amplitude of the impulse response in the top panel of 2C looks identical for different frequencies but then in the color matrix representation one can clearly see a high pass filtering. This needs an explanation here.

5. Results. L 513-520. To better describe the structure of the speech cochleogram, I would annotate the figure to clearly show that the first bands of energy harmonics that are due to the voice pitch, that the ones found in the middle are the formants and that there are then vertical bands in the higher frequencies that correspond once again to voicing (as you mention in the text). Also your text description should mention formants. Personally, I find them more visible in the spectrograms with df=10 or 30 while they appear compressed in a small range in the cochleogram.

6. Related to 5. In the discussion on the role of temporal vs spectral modulation (~l788-800) you focus on the detection of pitch. Here again one might want to talk about speech formants as well.

Reviewer #2: The authors compare multiple representations of a set of natural sounds. One of the representations is matched to encoding properties of the auditory processing pathway, as inferred from neural and perceptual experiments. In particular, it makes use of a two stage filter in which each stage uses bandwidth scaled filters (the filter bandwidth increases with the center-frequency). The other representation is standard in audio signal processing techniques and uses constant bandwidth filters in both stages. The authors demonstrate that the biologically inspired "bandwidth-scaled" representation is more efficient (i.e., higher entropy representations of natural sounds) than the standard audio representation.

Although, it is not a new idea that perceptual systems are optimized to efficiently represent natural stimuli, this work is a novel and worthy contribution to that field of work. The logic of the paper seems sound, the methods are appropriate, and the results support the conclusions. Given this, I believe this paper is of interest to the scientific community and merits publication.

I find no major flaws and I will devote the rest of this review to suggestions and comments for the authors to consider. Three of them I think are quite important. The rest are minor.

+ Important comment 01: It was only on reaching the paper discussion that I realized I wasn't sure what the authors meant by "bandwidth scaling". Given its prominent usage in the title and abstract I think it should be defined more clearly early in the paper and continually emphasized throughout.

In the abstract it is stated that: "bandwidth scaling produces a frequency-dependent gain that counteracts the tendency of natural sound power to decrease with frequency, resulting in a whitened output representation." I find this phrasing confusing. The "gain" results from the "larger bandwidth" ppoling signal over a larger range of frequencies. This is more than just a gain as it extracts different features from the audio waveform.

In addition, the title of the paper references "two stages of bandwidth scaling" but the definition in the abstract only references the first stage.

I assume the title refers to the fact that in both stages the filter bandwidths grow larger as the center-frequency increases. This should be stated explicitly. If the term means something more specific than what I have just mentioned, that should be stated. Either way the definition should be prominent in the paper and should clearly reference both stages of the processing heirarchy. Otherwise the title should be modified.

+ Important comment 02: In both the abstract (lines 30-32: ) and in the discussion (lines 792-793) the authors state that cochleagrams "sacrifice spectral information while producing a more robust temporal representation" or "accentuate temporal information" relative to short-time Fourier Transform (STFT). I find this phrasing confusing. A STFT can easily accentuate a temporal information by using shorter windows. There is nothing inherent in an STFT that favors spectral resolution over temporal resolution. I see the major difference between the two as: STFT uses the /same/ spectrotemporal resolution for all frequencies (arbitrarily so); whereas the cochleagram representation favors better spectral resolution in low-frequencies and better temporal resolution at high-frequencies (for good reasons, as these results show). Although, this is a more subtle distinction, I believe it is more accurate, and I suggest the authors state something like this in both the abstract and discussion. A clear and succinct definition of bandwidth-scaling will aid this, as it is precisely the bandwidth scaling that cochleagrams (and midbrain modulation spectral representations) have that Fourier Transforms and conventional modulation power spectra lack.

+ Important comment 03: the authors state in the submission that code for their auditory model is available via GitHub. I searched both the main text and supplemental for a URL but did not find one. A link to the code repository will be very helpful.

+ minor comment: the authors have not demonstrated that the biologically inspired representation is an "optimally efficient" representation, only that it is more efficient than more broadly known representations (Short-Time Fourier Transform and Modulation Power Spectrum). I would be curious to know how the representation entropy changed with more subtle changes in representation (e.g. what if the empirically determined constants in Equations (3), (7) etc. were altered to create a /different form of bandwidth scaling/? What if the number of cochlear bins were altered? How does the sparse coding representation Lewicki [Ref 23] compare?). This is not a small request, and a full answer perhaps deserves a separate paper. But perhaps this question could be raised in the discussion?

+ minor comment: another issue that would be of interest to many readers is how this compares to studies of other perceptual systems. Is it known that the visual or olfactory system use such "whitened" representations. Is an analogue of bandwidth scaling involved? If so, pointing readers to the relevant papers would be helpful. If not, the authors might remark upon this.

+ minor suggestion: Table S1 would be more interesting if it included columns for the entropies of each representation. I would be curious to see which sounds are outliers in this representation scheme. Another way to show this could be to simply print the cochleagram/spectrogram entropy value by each subplot in Fig 4, and the midbrain/MPS entropy value by each subbplot in Fig 7.

+ minor suggestion: Fig 7s A-C all use the same colorscale, which means figs 7C look boring to the eye. This clearly makes the point that the midbrain inspired representation is "whitened" relative to the modulation power spectrum. But it would be interesting to see these plots on an adjusted colorscale that shows their structure more clearly. Perhaps a supplementary figure? Or an additional subplot 7D?

+ minor suggestion: Line 208 ("Similarly, perceptually measured modulation bandwidths in human listeners scale with modulation frequency [25,26].") I find confusing. The modulation bandwidths of the auditory system are /inferred/ from perceptual thresholds (it is these that are /measured/).

+ minor suggestion: at Line 244, I expected to read about how Modulation Power Spectra were computed for the Fourier spectrographic decompositions. This would mirror the earlier structure describing how the biologically inspired heirarchical representations were constructed. I would suggest moving the section on "Spectro-temporal resolution and uncertainty principle" to after the section on "Modulation power spectrum (MPS)".

+ minor suggestion: Line 280. I was initially confused by what was meant by the "two biologically inspired sound decompositions" and had to read several times before I realized the two levels of the heirarchical decomposition were being discussed as two separate decompositions. I would suggest the authors be mindful of this and re-word the paragraph.

+ minor suggestion: Lines 286-289: I don't think the technical details of how Singh and Theunissen computed the Fourier MPS relevant. This is complexity that will tax a reader but doesn't help them understand what is being done here. I would suggest removing these sentences and cutting straight to Eq ([Disp-formula pcbi.1010862.e022]). A simple citation of Singh and Theunissen here will give them due credit.

+ minor suggestion: lines 432-433: "In practice, the selection of these ranges has a minimal impact on the entropy calculation and does not account for the entropy differences between sounds.". This is too technical for the main text but some readers might appreciate a supplementary figure showing this.

**Have the authors made all data and (if applicable) computational code underlying the findings in their manuscript fully available?**

Reviewer #1: Yes

Reviewer #2: Yes

PLOS authors have the option to publish the peer review history of their article (what does this mean?). If published, this will include your full peer review and any attached files.

Reviewer #1: No

Reviewer #2: No
---

## [Decision Letter · Decision Letter 1]

27 Sep 2022

Dear Dr Escabi,

Thank you very much for submitting your manuscript "Two stages of bandwidth scaling drives efficient neural coding of natural sounds" for consideration at PLOS Computational Biology.

Your revised manuscript was reviewed by members of the editorial board and by two independent reviewers. In light of the reviews (below this email), we would like to invite the resubmission of a revised version that addresses the set of issues raised by Reviewer # 1.

We cannot make any decision about publication until we have seen the revised manuscript and your response to the reviewers' comments. Your revised manuscript is also likely to be sent to reviewers for further evaluation.

Sincerely,

Xuexin Wei

Academic Editor

PLOS Computational Biology

Samuel Gershman

Section Editor

PLOS Computational Biology

Reviewer's Responses to Questions

**Comments to the Authors:**

Reviewer #1: Dear authors,

Thank you for your detailed answers to my first round of reviews. I am mostly satisfied and I am glad that your manuscript now also mentions the density of hair cells in the cochlea and the fact that it is not perfectly logarithmic at low frequencies but a bit more linear (at least in humans). However, I believe that you did not completely understand my point relating to the power gain – and I was probably not completely clear. I am not disagreeing that there is a whitening in the frequency power curves (and as you describe well this is on average, etc). Clearly this is true because higher frequency auditory fibers have larger tuning bandwidth and thus, they integrate over a larger frequency range. Since (on average) natural sounds have less energy in the high-frequency range, the result is a more equal distribution of power response per unit. And more uniform distribution yields a higher entropy. This is the main message of the first part of the paper and it is well done and convincing. In addition, you perform this calculation for two levels of processing by including the modulation tuning; I very much appreciate that effort. It yields a nice and complete picture.

However, I think that one has to be more careful when talking about power, gain and the slopes (your analyses in figures 4 and 5). More precisely, a frequency power spectrum has units of density: power per frequency. One can plot that power density using linear or log units in the x axis. When one does a cochleogram and then estimates power, you get an “equal sampling” in log units – so now your power density curve is in power per octave. You could still plot that in linear or log frequency units but the curve means something else that the power per frequency. For example, when you compare power in Fourier Spectral and Cochlear Spectral, you can either compare two different densities in power/f vs power/logf units as you have done in figure 4 or attempt to compare densities in the same units. If you did this by transforming the power/logf obtained with the cochlear filter into a power/f, you would get a boost in the low frequencies. Think of your 100 cochlear filters distributed in log scale (as in 100 red dots in Fig.4) – now you resample these into 100 bins in linear scale by summing or splitting the energy as needed (but not by taking the average). You will then obtain a density in power/f. And you will also see that the low f are boosted once again. I think that this difference explains the discrepancy between the “whitening” that maximizes the “lifetime” spectral entropy across units and the experiments in loudness, frequency discrimination, etc that suggest the oversampling in the lower frequency range. And I also think that for the power/gain argrument, it only makes sense to compare curves that have the same units. You could then clearly distinguish the entropy argument (a uniform sampling) with the power argument. If you were able to clearly explain this in your paper, I think that it would have a greater impact.

I am happy to discuss this point with you so that we can reach an agreement. (Alternatively, I could write a comment to your paper).

Minor points.

1. As I was rereading your paper, I wonder to what extent equations 22 and 23 depend on N. Clearly if the numerator of 23 goes as logN then the exact number N does not matter. I am suspecting that as N gets large enough this is the case. Did you check whether you were in that regime?

2. In Fig 5B, I would have axis labels at 0.5 and 1.0 instead of 0.6 and 1.1. 1.0 is the maximum value of these scaled entropy and it makes more sense to be very explicit about that.

Reviewer #2: I am satisfied the authors have addressed all issues I raised.

**Have the authors made all data and (if applicable) computational code underlying the findings in their manuscript fully available?**

Reviewer #1: Yes

Reviewer #2: Yes

PLOS authors have the option to publish the peer review history of their article (what does this mean?). If published, this will include your full peer review and any attached files.

Reviewer #1: **Yes: **Frederic Theunissen

Reviewer #2: No
---

## [Decision Letter · Decision Letter 2]

9 Jan 2023

Dear Dr Escabi,

We are pleased to inform you that your manuscript 'Two stages of bandwidth scaling drives efficient neural coding of natural sounds' has been provisionally accepted for publication in PLOS Computational Biology.

Best regards,

Xuexin Wei

Academic Editor

PLOS Computational Biology

Samuel Gershman

Section Editor

PLOS Computational Biology

Reviewer's Responses to Questions

**Comments to the Authors:**

Reviewer #1: Thanks for making those changes. I think that I would have still worded the results differently and stressed more the meaning of Sup Fig. 5 but it is all there.

Congrats on a nice study.

Frederic.

**Have the authors made all data and (if applicable) computational code underlying the findings in their manuscript fully available?**

Reviewer #1: None

PLOS authors have the option to publish the peer review history of their article (what does this mean?). If published, this will include your full peer review and any attached files.

Reviewer #1: **Yes: **Frederic Theunissen

---

## [Editor Report · Acceptance letter]

2 Feb 2023

PCOMPBIOL-D-22-00664R2 

Two stages of bandwidth scaling drives efficient neural coding of natural sounds

Dear Dr Escabí,

I am pleased to inform you that your manuscript has been formally accepted for publication in PLOS Computational Biology. Your manuscript is now with our production department and you will be notified of the publication date in due course.

With kind regards,

Timea Kemeri-Szekernyes
